# Generative Causal Structure Learning with Dual Latent Spaces and Annealing

**Soma Bandyopadhyay**                          *soma.bandyopadhyay@tcs.com*
*Department of Artificial Intelligence*
*Indian Institute of Technology Kharagpur*
*TCS Research, TATA Consultancy Services Limited*

**Sudeshna Sarkar**                              *sudeshna@cse.iitkgp.ac.in*
*Department of Computer Science and Engineering*
*Indian Institute of Technology Kharagpur*

**Reviewed on OpenReview:** *https://openreview.net/pdf?id=wI5rFWfjKV*

## Abstract

In this work, we address causal structure learning in the presence of unobserved confounders. Such causal structures can be represented by Acyclic Directed Mixed Graphs (ADMGs), where observed cause-effect relations are depicted by directed edges and unobserved confounded relations by bidirected edges. Prior methods for causal structure learning with unobserved common causes have primarily focused on search-based approaches, and more recently on flow-based generative models. We propose a novel generative method based on a variant of the Variational Autoencoder (VAE) with dual latent spaces to represent the directed cause-effect relations and the bidirected unobserved confounded relations, associating two trainable adjacency matrices. To enhance the learning process, we introduce a causality constraint combined with the concept of a causal annealing strategy during training, guiding the learning toward meaningful causal structures. Experimental results show that our method achieves competitive performance in identifying both observed and latent causal relationships on synthetic datasets. Furthermore, we demonstrate that the learned causal structure significantly improves downstream causal inference performance on real-world data. [1]

## 1 Introduction

Learning cause–effect relationships without known causal structures is fundamental to causal reasoning. In many real-world scenarios, unobserved common causes distort statistical associations among features, making structure learning more challenging. For example, in human mobility, external factors such as weather can jointly influence departure time and destination, acting as latent confounders that bias inferred relations.

This work aims to learn causal structures among features in the presence of unobserved (latent) confounders, without relying on prior structural knowledge. Further, we demonstrate the applicability of the proposed method for the causal inference task on a real-world dataset, leveraging the learned causal relationships.

Prior studies on causal structure identification under unobserved confounding have predominantly employed score-based approaches, using criteria such as the Bayesian Information Criterion (BIC), and constraint-based methods relying on conditional independence tests.

Subsequently, the causal structure discovery has been reformulated as a continuous optimization problem enforcing acyclicity constraints (Zheng et al., 2018) via differentiable function with gradient-based techniques,

---

[1]Code available at: OSF repository

excluding the need for a combinatorial search. However, these continuous optimization based methods assume causal sufficiency and do not consider unobserved variables.

Recently, the **bow-free** constraint (Bhattacharya et al., 2021) applied to Acyclic Directed Mixed Graphs (ADMGs) (Richardson & Spirtes, 2002). ADMG has been utilized to represent both direct causal relationships, and unobserved confounding relations. The observed cause-effect relations are represented by directed edges. The unobserved confounders are represented by bidirected edges between the variables which have unobserved common cause that influences both variables. The bow-free constraint prevents overlapping directed and bidirected edges between the same pair of variables. Specifically, if there is a directed edge $V_i \to V_j$, then there cannot also be a bidirected edge $V_i \leftrightarrow V_j$. This concept, integrated into Structural Causal Models (SCMs) (Pearl, 2009), has recently been explored in neural ADMG contexts.

We follow ADMG structure estimation assumptions with latent confounding from prior work (Ashman et al., 2023). The framework assumes a nonlinear additive-noise SCMs, where each variable is modelled as a nonlinear function of its direct causes combined with independent noise and exogenous latent confounders. These assumptions support estimation of directed and bidirected edges from observational data, however observationally equivalent structures may still arise.While these assumptions enable plausible causal interpretations, they do not guarantee exact recovery of the ground-truth SCM, and a discussion on causal guarantees is provided in Appendix D.

We represent the directed cause–effect relationship by the directed adjacency matrix $A^{(\text{Observed})}$ or $A_D$, which captures **directed, asymmetric, observed cause-effect** relationships. The unobserved common cause association with different pair of variables is represented by the bidirected adjacency matrix $A^{(\text{Unobserved})}$ or $A_B$ which captures **bidirected, symmetric, unobserved confounded** associations.

Our approach employs a Variational Auto Encoder (VAE) (Doersch, 2021; Kingma & Welling, 2013) based **causally constrained** generative framework designed to estimate unobserved causal structures within the ADMG formalism. We denote this model as **G-ADMG-CL** (Generative Acyclic Directed Mixed Graph based Causal Structure Learning). Detection of unobserved confounders is enabled under the bow-free ADMG assumption (Ashman et al., 2023), using a nonlinear additive-noise SCM. The framework captures both nonlinear cause–effect relationships and latent confounding through **trainable adjacency matrices $\mathbf{A_D}$ and $\mathbf{A_B}$**.Overall, the learned adjacency matrices provide interpretable causal representations that jointly encode observed and latent relationships, enabling structure discovery under confounding. The proposed framework is characterized by the following components:

- A VAE-based architecture that disentangles observed and unobserved causal relations through **dual latent spaces**.

- Separate latent spaces are dedicated to directed cause–effect relationships and unobserved confounded relations, governed by $A_D$ and $A_B$, respectively.

- A **causally aware objective function** that enforces acyclicity for observed relations, bow-free constraints for confounded edges, and entropy-sparsity regularization to balance structural variability while preserving the asymmetric/symmetric nature of $A_D$ and $A_B$.

- A **causal annealing** strategy that gradually increases the causal-regularization weight from 0 to 1 until the **causal transition epoch (CTE)**, prioritizing reconstruction and KL-divergence optimization during early training before activating full causal constraints.

We evaluate G-ADMG-CL on synthetic data generated from nonlinear SCMs and real-world data demonstrating effective causal structure learning under latent confounding.

## 2 Related Work

Causal structure learning is a fundamental problem with wide applications across scientific and real-world domains. Estimating causal relationships in the presence of *unobserved confounders* is both more realistic and more challenging. We review key existing approaches, with a particular focus on methods that attempt to model latent confounding and capture both directed and bidirected dependencies.

**Score-based and constraint-based methods:** Classical approaches to causal discovery (Hasan et al., 2023; Zanga et al., 2023) include constraint-based and score-based methods. Constraint-based methods, such as the PC and FCI algorithms (Spirtes et al., 2000), rely on conditional independence testing to infer causal graphs, while score-based methods search for graph structures that optimize predefined scoring criteria such as the BIC. However, these methods typically assume access to a fully observed variable set or struggle with indistinguishability in equivalence classes under hidden confounding.

**Differentiable DAG learning (no latent confounding):** The NOTEARS framework (Zheng et al., 2018) introduced a differentiable acyclicity constraint based on the trace of a matrix exponential, enabling gradient-based optimization for DAG structure learning. This approach has been extended by neural methods such as DAG-GNN (Yu et al., 2019b) and N-DAG-G (Geffner et al., 2022), which integrate deep architectures for end-to-end causal discovery. While effective for fully observed systems, these methods assume causal sufficiency and cannot represent latent confounding or bidirected edges. (Zecevic et al., 2021) relate graph neural networks to structural causal models on graph-structured data, offering a causal interpretation of message passing but without explicit modelling of latent confounders.

**Handling unobserved confounders:** Several methods explicitly attempt to model latent confounding. Fast Causal Inference (FCI) (Spirtes et al., 2000) is a constraint-based method capable of detecting latent variables through conditional independence tests, but it cannot distinguish between equivalence classes with identical independencies. Repetitive Causal Discovery (RCD) (Maeda & Shimizu, 2020) handles linear non-Gaussian models and introduces bidirected edges to represent unobserved common causes. CAM-UV (Maeda & Shimizu, 2021) extends the Causal Additive Model (CAM) to latent-variable settings by using HSIC-based independence tests (Gretton et al., 2007) combined with a scoring procedure. More recently, Gonzales & Valizadeh (2024) proposed a score-based global search over augmented DAGs that can handle latent confounding, but is restricted in scalability.

**Ancestral graphs and bow-free constraints:** To more explicitly model both directed and bidirected edges under latent confounding, recent works have extended the causal discovery framework to ADMGs. The approach in (Bhattacharya et al., 2021) introduces a differentiable objective incorporating acyclicity, ancestral constraints, a c-tree penalty, and the bow-free constraint, targeting linear Gaussian additive noise models. These constraints help ensure identifiability and interpretability in the learned graphs. None of these methods utilizes a generative model.

**Flow-based methods:** Despite these advances, most existing methods are either search-based or rely on structural constraints but do not leverage a generative modelling perspective. A notable exception is (Ashman et al., 2023), which introduces a flow-based auto-regressive model for estimating ADMG structures with nonlinear relationships. Their method supports both bow-free (N-BF-ADMG-G) and general ADMG settings (N-ADMG-G), and is capable of modelling latent confounders. However, it lacks latent-variable disentanglement and does not optimize for structure recovery through generation-based objectives.

## 3 Methodology

**Task overview:** In this section, we elaborate the two tasks. First, **causal structure learning** in the presence of both observed cause–effect and unobserved confounded relations, using the proposed **G-ADMG-CL** model. Second, we extend this framework to **prediction and causal inference** (denoted G-ADMG-CL+P), where the learned causal graph supports downstream estimation of treatment effects, such as the Average Treatment Effect (ATE). This extension integrates the inferred structural knowledge into the decoder to enable structure-aware prediction and causal inference.

**Method overview:** We propose **G-ADMG-CL**, a *causality-constrained VAE* to jointly learn both directed and bidirected causal relationships in the presence of latent confounding. The key innovations in our method are the introduction of dual latent spaces, causality constraint optimization, and a causal annealing strategy to guide causality learning. Our approach disentangles observed cause–effect interactions from unobserved confounded associations using these dual latent spaces. To guide structure learning, we propose a causality-aware objective that incorporates acyclicity, bow-free constraints, and structural regularization. We introduce a novel *causal annealing* strategy that gradually enforces causal constraints during training, enabling more stable and accurate graph recovery. The goal of our method is to learn both edge types simultaneously

while ensuring that the resulting structure supports robust ADMG structure estimation, is interpretable, and aligned with the true underlying causal mechanisms.

**Causal structure learning:** VAE is a popular generative model for learning disentangled latent representation (Higgins et al., 2017). Its probabilistic formulation allows incorporating constraints in the latent space while learning and optimizing the loss function. The functional components of this method are illustrated in Figure 1.

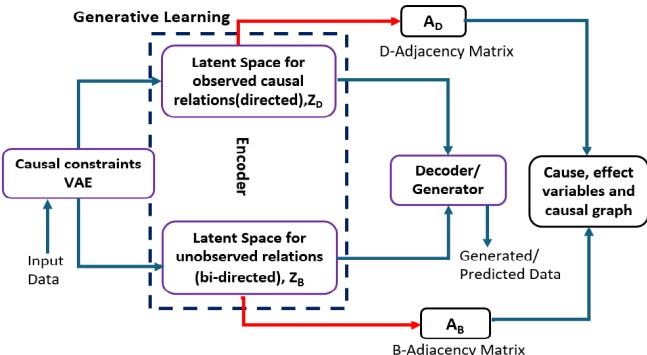

Figure 1: Functional components of the proposed model, featuring dual latent spaces $Z_D$ and $Z_B$ guided by the trainable adjacency matrices $A_D$ and $A_B$ for causal structure learning.

**Functional components:** The encoder maps the input data into two disentangled latent spaces: the directed latent space $Z_D$ representing observed causal relations, and the bidirected latent space $Z_B$ representing unobserved confounded relations. These latent representations are used to learn two corresponding trainable adjacency matrices, $A_D$ and $A_B$. Causal constraints are incorporated during training to guide the learning process to ensure meaningful structural recovery. The learned adjacency matrices are fed into the decoder/generator to reconstruct or generate the predicted data, and to obtain the inferred causal structure, integrating both observed cause-effect and unobserved confounded relations. The predicted data is further utilized for causal inference tasks. Additionally, the proposed causal annealing approach further guides the causal structure learning to enforce causal constraints during training.

**Detailed methodology:**

**Encoder:** The encoder uses dual latent spaces to learn the observed and unobserved cause–effect relationships. The directed latent space $z_{\text{directed}} (\equiv z_D)$ learns the observed causal relationship, and the bidirected latent space $z_{\text{bidirected}} (\equiv z_B)$ addresses unobserved or confounded relationship. Latent space representations for both latent spaces are obtained by computing mean $\mu_{\text{directed}} (\equiv \mu_D)$, $\mu_{\text{bidirected}} (\equiv \mu_B)$, and log-variance $\log \sigma^2_{\text{directed}}$, $\log \sigma^2_{\text{bidirected}}$ respectively. Latent variables are sampled using the reparameterization trick: first $\epsilon$ is element-wise sampled from $\mathcal{N}(0,1)$ and then are computed using Eq. 2. We compute $\mu_{\text{directed\_guided\_Adjacency}} (\equiv \mu_{DA_D})$ and $\mu_{\text{bidirected\_guided\_Adjacency}} (\equiv \mu_{BA_B})$ using Eq. 1 to guide $z_D$ and $z_B$ using the directed cause–effect relationships encoded in $A_D$ and the unobserved confounded relationships encoded in $A_B$. The aim is to obtain the causally structured latent variables, improving reconstruction and making the trained model more interpretable by following the cause–effect relationships.

The encoder comprises two fully connected layers (128 and 64 units) with ELU activations; dropout is applied for regularization.

$$\mu_{DA_D} = \mu_{\text{directed}} \cdot \mathbf{A}_D, \quad \mu_{BA_B} = \mu_{\text{bidirected}} \cdot \mathbf{A}_B. \tag{1}$$

$$z_D = \mu_{DA_D} + \epsilon_D \odot \exp\left(0.5 \cdot \log \sigma^2_D\right), \quad z_B = \mu_{BA_B} + \epsilon_B \odot \exp\left(0.5 \cdot \log \sigma^2_B\right), \tag{2}$$

**Decoder:** The decoder combines the **causally guided dual latent spaces** $[z_{\mathbf{D}}, z_{\mathbf{B}}]$ from the encoder and reconstructs the given input from the combined latent spaces, retaining the causal and unobserved confounded relationship among the variables. This reconstructed outcome is further exploited for the **prediction task**. The decoder consists of two fully connected layers with 64 and 128 units, respectively, using LeakyReLU activations and dropout for regularization.

**Generator:** The decoder is employed to generate new data samples by passing the latent samples sampled from a Gaussian distribution.

**Loss function ($\mathcal{L}_{\mathbf{total}}$)** (Eq. (3)): This comprises three components. The first two components are already well-studied in VAE-based models. We introduce a novel third component, the **causal mixed graph loss** $\left(\mathcal{L}_{\text{Causal\_ADMG}}\right)$ weighted by the $\lambda_{\text{causal}}$.

The loss function follows the structure of an ADMG. The adjacency matrices $A_D$ and $A_B$ are learned end-to-end and are parameterized via trainable weight matrices $W_1$ and $W_2$, respectively. Using this causality constraint, $\left(\mathcal{L}_{\text{Causal\_ADMG}}\right)$ the training process enforces the structural regularization aligned properties of ADMGs, and learn causal relationships in the presence of unobserved confounders.

**Total Loss:**

$$\mathcal{L}_{\text{total}} = \mathcal{L}_{\text{reconstruction}} + \lambda_{\text{KL}}\left(\mathcal{L}_{\text{KL\_directed}} + \mathcal{L}_{\text{KL\_bidirected}}\right)$$
$$+ \lambda_{\text{causal}}\left(\mathcal{L}_{\text{Causal\_ADMG}}\right) \tag{3}$$

1. **Reconstruction loss:** $\mathcal{L}_{\text{reconstruction}} = \|\mathbf{x} - \hat{\mathbf{x}}\|_2^2$, the mean-squared error (MSE) between input (x) and reconstructed data ($\hat{x}$), standard for Gaussian VAEs.

2. **KL divergence loss:** This is for latent space regularization, where, $\mathcal{L}_{\text{KL\_directed}}$ (Eq. (4)) for the directed, and $\mathcal{L}_{\text{KL\_bidirected}}$ (Eq. (5)) for the bidirected KL divergence. We apply KL annealing (Fu et al., 2019), kl_weight ($\lambda_{\text{KL}}$) (a gradually increasing weight) to multiply the KL divergence term to counter KL-vanishing during the initial training phase and focus primarily on minimizing the reconstruction error.

$$\lambda_{\text{KL}} \cdot L_{\text{KL\_directed}} = \lambda_{\text{KL}} \cdot \text{KL}\left(q_\phi(\mathbf{z}_{\text{directed}}|\mathbf{X}) \parallel p_\theta(\mathbf{z}_{\text{directed}}|\mathbf{X})\right), \tag{4}$$
$$\lambda_{\text{KL}} \cdot L_{\text{KL\_bidirected}} = \lambda_{\text{KL}} \cdot \text{KL}\left(q_\phi(\mathbf{z}_{\text{bidirected}}|\mathbf{X}) \parallel p_\theta(\mathbf{z}_{\text{bidirected}}|\mathbf{X})\right) \tag{5}$$

3. **Causal mixed graph loss** ($\mathcal{L}_{\text{Causal\_ADMG}}$): This is for causal structure regularization, with the causal regularization weight $\lambda_{\text{causal}}$ as shown in Eq. (6), comprises the following components:

$$\mathcal{L}_{\text{Causal\_ADMG}} = \lambda_{\text{cycle}}\mathcal{L}_{\text{cycle}}(A_D) + \lambda_{\text{bow}}\mathcal{L}_{\text{bow}}(A_D, A_B)$$
$$+ \lambda_{\text{entropy}}(A_D)\mathcal{L}_{\text{entropy}}(A_D) + \lambda_{\text{entropy}}(A_B)\mathcal{L}_{\text{entropy}}(A_B)$$
$$+ \lambda_{\text{asymmetry}}(A_D)\mathcal{L}_{\text{asymmetry}}(A_D) + \lambda_{\text{symmetry}}(A_B)\mathcal{L}_{\text{symmetry}}(A_B)$$
$$+ \lambda_{\text{sparsity}}(A_D)\mathcal{L}_{\text{sparsity}}(A_D) + \lambda_{\text{sparsity}}(A_B)\mathcal{L}_{\text{sparsity}}(A_B). \tag{6}$$

- **Acyclicity:** $\mathcal{L}_{\text{cycle}}$ (Eq. (7)) applies acyclic constraints on $A_D$ to prevent cyclic causal structures among directed latent variables. Minimizing $\mathcal{L}_{\text{cycle}}$ enforces acyclicity by penalizing non-zero $\text{trace}(e^{A_D})$.
- **Bow-free:** $\mathcal{L}_{\text{bow}}$ (Eq. (8)) penalizes simultaneous directed and bidirected edges, restricting variable pairs from sharing both cause-effect and unobserved confounded relations. $\mathcal{L}_{\text{bow}}(A_D, A_B)$ ensures the bow-free property by suppressing such dual connections.
- **Entropy:** $\mathcal{L}_{\text{entropy}}(A_D)$ and $\mathcal{L}_{\text{entropy}}(A_B)$ (Eq. (9)) maintain variability in latent spaces to enforce steady learning.
- **Symmetry–Asymmetry:** $\mathcal{L}_{\text{asymmetry}}(A_D)$ ensures non-existence of both $A_D[i,j]$ and $A_D[j,i]$, while $\mathcal{L}_{\text{symmetry}}(A_B)$ ensures simultaneous existence of $A_B[i,j]$ and $A_B[j,i]$, as $A_D$ is asymmetric and bidirected edges in $A_B$ are symmetric (Eq. (10)).
- **Sparsity:** Reduces noise in both directed $A_D$ and bidirected $A_B$ adjacency matrices (Eq. (11)).

$$\mathcal{L}_{\text{cycle}} = \left|\text{trace}(e^{A_D}) - d\right|, (d : dimension A_D, A_B) \tag{7}$$

$$\mathcal{L}_{\text{bow}} = \alpha \cdot \left(\frac{1}{n}\sum_{i,j} A_D[i,j]^2\right) \cdot \beta \cdot \left(\frac{1}{n}\sum_{i,j} A_B[i,j]^2\right) \tag{8}$$

$$\mathcal{L}_{\text{entropy}}(A_D) = -\sum_i p_i^{(A_D)} \log p_i^{(A_D)}, \quad \mathcal{L}_{\text{entropy}}(A_B) = -\sum_i p_i^{(A_B)} \log p_i^{(A_B)}$$

$$p_i^{(A_D)} = \frac{|A_D[i]|}{\sum_j |A_D[j]|}, \quad p_i^{(A_B)} = \frac{|A_B[i]|}{\sum_j |A_B[j]|}$$

(9)

$$\mathcal{L}_{\text{asymmetry}}(A_D) = \|A_D \odot A_D^\top\|_1, \quad \mathcal{L}_{\text{symmetry}}(A_B) = \|A_B - A_B^\top\|_F^2 \tag{10}$$

$$\mathcal{L}_{\text{sparsity}}(A_D) = \|A_D\|_1, \quad \mathcal{L}_{\text{sparsity}}(A_B) = \|A_B\|_1 \tag{11}$$

**Causal annealing:** We introduce **causal annealing**, a learning strategy designed to systematically control the influence of the causal regularisation within the total loss during training, where we keep $\lambda_{\text{causal}}$ a gradually increasing between 0 to 1 until the **causal transition epoch (CTE)**. At CTE $\lambda_{\text{causal}}$ becomes 1 and remains 1 for the rest of the learning cycle. It helps to learn the data characteristics associated with the reconstruction and KL divergence components, and then apply the causality constraints. Algorithm 2 describes the causal annealing. The proposed causal annealing has two schedules, one is the `default hard mode`, where $\lambda_{\text{causal}}$ remains 0 for $e < CTE$, and switches to 1 at CTE. To avoid abrupt changes in the gradient behavior and prevent abrupt instability at transition a relaxed schedule is introduced. In this relaxed schedule, a relaxed transition epoch $e_t$ is defined, during which $\lambda_{\text{causal}}$ starts to linearly increase from 0 at $e_t$ to 1 at the CTE.

---

**Algorithm 1** Causal Relationships Learning: G-ADMG-CL

---

1: **Input:** Data: $\mathbf{X} \in \mathbb{R}^{n \times d}$ with unobserved confounder
2: **Output:** Reconstructed data $\hat{\mathbf{X}}$, Directed ($A_D$), Bidirected ($A_B$) adjacency matrices
   — **Encoder Block** —
3: Initialize trainable adjacency matrices: (Once at start of training)
4: Directed adjacency matrix: $W_1 \sim \mathcal{N}(0,1)^{d \times d}$
5: Bidirected adjacency matrix: $W_2 \sim \mathcal{N}(0,1)^{d \times d}$
6: Compute directed latent statistics:
7: $\quad \boldsymbol{\mu}_D, \log \boldsymbol{\sigma}_D^2 \leftarrow$ FC layers on $\mathbf{X}$
8: $\quad \mu_{DA_D} \leftarrow \boldsymbol{\mu}_D \cdot W1$
9: Compute bidirected latent statistics:
10: $\quad \boldsymbol{\mu}_B, \log \boldsymbol{\sigma}_B^2 \leftarrow$ FC layers on $\mathbf{X}$
11: $\quad \mu_{BA_B} \leftarrow \boldsymbol{\mu}_B \cdot W2$
   — **Reparameterization** —
12: Sample structure aware latent variables using reparameterization:
13: $\quad \mathbf{z}_D \leftarrow \mu_{DA_D} + \boldsymbol{\epsilon}_D \odot \exp(0.5 \cdot \log \boldsymbol{\sigma}_D^2), \quad \boldsymbol{\epsilon}_D \sim \mathcal{N}(0,1)$
14: $\quad \mathbf{z}_B \leftarrow \mu_{BA_B} + \boldsymbol{\epsilon}_B \odot \exp(0.5 \cdot \log \boldsymbol{\sigma}_B^2), \quad \boldsymbol{\epsilon}_B \sim \mathcal{N}(0,1)$
15: Concatenate latent representations: $\mathbf{z} \leftarrow [\mathbf{z}_D, \mathbf{z}_B]$
   — **Decoder Block** —
16: $\quad \hat{\mathbf{X}} \leftarrow \text{Decoder}(\mathbf{z})$
   — **Adjacency Matrix Estimation (Per-epoch latent-to-adjacency decoding via $W_1$, $W_2$)**—
17: $\quad A_D \leftarrow f(\mathbf{z}_D) = \mathbf{z}_D W_1, \quad A_B \leftarrow f(\mathbf{z}_B) = \mathbf{z}_B W_2$
   — **Learning and Optimization** —
18: Optimize with causal constraints: as defined in Eq. (3) during training
19: $\mathcal{L}_{\text{total}} = \mathcal{L}_{\text{reconstruction}} + \lambda_{\text{KL}}(\mathcal{L}_{\text{KL\_directed}} + \mathcal{L}_{\text{KL\_bidirected}}) + \lambda_{\text{causal}}\mathcal{L}_{\text{Causal\_ADMG}}$
   — **Learned Causal Structure** —
20: $\quad$ Return $A_D$, $A_B$ for evaluation and $\hat{\mathbf{X}}$

---

Algorithm 1 presents G-ADMG-CL, which learns causal structures with unobserved confounders based on ADMG, without assuming a known causal graph. The objective function enforces acyclicity and the bow-free property, ensuring $A_D$ remains asymmetric and $A_B$ symmetric, while balancing randomness and sparsity. This method supports applications such as conditional and counterfactual data generation, providing interpretable results through its dual latent spaces and learned adjacency matrices.

---

**Algorithm 2** Causal Annealing During Training

---

1: **Input:** Total epochs $E$, causal transition epoch $CTE$, linear transition start epoch $e_t$, anneal mode (`"hard"` or `"linear"`)
2: **Output:** Causal regularization schedule $\lambda_{\text{causal}}$ for each epoch
3: Initialize $\lambda_{\text{causal}} \leftarrow 0$
4: **for** epoch $e = 1$ to $E$ **do**
5:     **if** `anneal_mode == "hard"` **then**
6:         **if** $e < CTE$ **then**
7:             $\lambda_{\text{causal}} \leftarrow 0$
8:         **else**
9:             $\lambda_{\text{causal}} \leftarrow 1$
10:         **end if**
11:     **else**
12:         **if** $e < e_t$ **then**
13:             $\lambda_{\text{causal}} \leftarrow 0$
14:         **else if** $e < CTE$ **then**
15:             $\lambda_{\text{causal}} \leftarrow \dfrac{e - e_t}{CTE - e_t}$
16:         **else**
17:             $\lambda_{\text{causal}} \leftarrow 1$
18:         **end if**
19:     **end if**
20:     Update model parameters using $\lambda_{\text{causal}}$
21: **end for**

---

In contrast, other VAE-based causal models, such as CausalVAE (Yang et al., 2021), learn disentangled causal representations consistent with a prespecified causal graph under the assumption of causal sufficiency, but they do not perform causal structure discovery or model latent confounding via bidirected edges. (Leeb et al., 2021) study how to probe causal relations in the latent space of auto-encoders using interventional assays, focusing on analyzing and manipulating learned representations rather than recovering an ADMG. Notably, none of these approaches employ a *causal annealing* strategy like ours.

**Causal Inference:** We further employ our method to the prediction task as G-ADMG-CL+P to conduct the causal inference using learned dual latent spaces, $[z_{\text{D}}, z_{\text{B}}]$. The learned directed cause-effect and unobserved confounded relationships from the latent spaces reduce the variability in the average treatment effect (ATE) (Eq. 12), the difference in the expected effect when the treatment or cause is applied depending on both observed and unobserved confounders.

$$\text{ATE} = \mathbb{E}[Y(1)] - \mathbb{E}[Y(0)], \tag{12}$$

where $Y(1)$, $Y(0)$ represent the outcomes with and without applying the treatment (or cause), respectively.

## 4 Data

To evaluate our method under varying levels of structural complexity and confounding we consider two synthetic datasets: Fork Collider (FC) and Erdős–Rényi (ER) and one real-world dataset IHDP for causal structure identification and causal inference tasks. Additionally, we conduct an early evaluation on the Sachs protein-signaling data (Sachs et al., 2005) elaborated in Appendix F.

**Fork Collider (FC):**

We synthesize 4000 samples of this data using the Eq. 13 of the SCM following (Ashman et al., 2023). The causal graph and adjacency matrices are depicted in Figure 2a. Variables X2, X3 and X3, X4 have bidirected edges, where u1 and u2 are the respective confounders influencing these pair of variables. The bidirected

edges are represented by the dotted line, and the directed edges are shown by the solid line.

$$
\begin{aligned}
\mathbf{T} &= [u_1, u_2, \epsilon_1, \epsilon_2, \epsilon_3, \epsilon_4, \epsilon_5]^T \sim \mathcal{N}(\mathbf{0}, \mathbf{I}), \\
x_1 &= \epsilon_1, \\
x_2 &= \sqrt{6}\exp(-u_1^2) + 0.1\epsilon_2, \\
x_3 &= \sqrt{6}\exp(-u_1^2) + \sqrt{6}\exp(-u_2^2) + 0.2\epsilon_3, \\
x_4 &= \sqrt{6}\exp(-u_2^2) + \sqrt{6}\exp(-x_1^2) + 0.1\epsilon_4, \\
x_5 &= \sqrt{6}\exp(-x_1^2) + 0.1\epsilon_5.
\end{aligned}
\tag{13}
$$

**Erdős–Rényi (ER):** We evaluate our method on two ER based synthetic graph configurations: ER(4,6,4) and ER(12,50,10), with unobserved confounders, where $(d, e, m)$ denote observed nodes, directed, and bidirected edges, respectively, each simulated with 4000 samples. The first configuration, ER(4,6,4), contains $d = 4$ variables, $e = 6$ directed edges, and $m = 4$ bidirected edges representing a compact graph with moderate confounding. The second, ER(12,50,10), includes $d = 12$ variables, $e = 50$ directed edges, and $m = 10$ bidirected edges, modelling a more complex and densely connected structure. These settings are designed to evaluate the scalability and robustness of our model in recovering both observed and latent causal relationships under increasing graph density and confounding.

Data are generated using the Eq. (18) (Appendix A), following the ER-based ADMG simulation procedure described in (Ashman et al., 2023). The corresponding causal graphs and adjacency matrices are illustrated in Figure 3a. Notably, in these ER settings, directed (solid) and bidirected (dotted) edges often overlap reflecting realistic scenarios where a pair of variables possessing direct cause–effect relationship may be influenced by the unobserved common cause.

**IHDP**: IHDP (Infant Health and Development Program) is a real-world dataset originally introduced by (Hill, 2011) comprising measurements of infants and their mothers, collected during a randomized experiment. We adopt the IHDP dataset (Louizos et al., 2017), for the causal inference task, where the goal is to estimate the effect of home visits by specialists (cause) on the cognitive development of infants (effect). The dataset is used with 10 replicates, each consisting of a 70% train (IHDP_train) and 30% test (IHDP_test) split, with a fixed size of 747 individuals per replicate. It includes 19 binary covariates and 6 continuous covariates (e.g., birth weight, head circumference), all normalized before training. To incorporate unobserved confounding, treated individuals with non-white mothers are excluded. Outcomes are simulated based on Setting B (log-linear response surfaces), and Gaussian exogenous noise is applied.

## 5 Results

We evaluate the proposed G-ADMG-CL model across both synthetic and real-world datasets to assess its ability to recover causal structure in the presence of latent confounders and to support causal inference.

### 5.1 Performance Measure

We evaluate model performance using the following metrics:

**F1 Score:** The harmonic mean of precision and recall, computed over predicted vs. ground-truth edges (directed and bidirected separately). It is defined as:

$$
\text{F1} = 2 \cdot \frac{\text{Precision} \cdot \text{Recall}}{\text{Precision} + \text{Recall}}
\tag{14}
$$

**RMSE-ATE:** Measures the error in causal inference by computing the root mean squared error between the predicted average treatment effect (ATE_P) and ground-truth ATE over $n$ samples:

$$
\text{RMSE-ATE} = \sqrt{\frac{1}{n}\sum_{i=1}^{n}(\text{ATE\_P}_i - \text{ATE}_i)^2}
\tag{15}
$$

## 5.2 Evaluation of G-ADMG-CL

We evaluate the performance of the learned causal structure across different datasets by comparing it with various established state-of-the-art methods. F1 scores are used to measure the performance. The $F1_D$, $F1_B$ represent the F1 scores obtained for directed cause-effect relationships and for the bidirected unobserved confounded relations among the variables, respectively.

We compare our method against the following state-of-the-art approaches:

- FCI: Constraint-based method using independence tests to detect confounders.

- CAM-UV: Additive noise model with HSIC tests for hidden variables.

- RCD: Learns linear non-Gaussian models with bidirected edges.

- DCD: Additive causal discovery with structural regularization.

- N-DAG-G: Neural DAG learner assuming no hidden confounding.

- N-ADMG-G: Flow-based ADMG learner capturing latent confounders.

- N-BF-ADMG-G: Flow-based model for ADMGs with bow-free constraints.

Unlike these approaches, our method leverages *dual latent space* learning with *causality constraints* to jointly model both directed and bidirected causal relationships. Additionally, we implement a causal annealing strategy that enhances performance and training stability, leading to more reliable estimation of latent confounding structures. Together, these innovations promote more effective disentanglement of causal directions and latent confounders, leading to consistent performance gains in both $F1_D$ and $F1_B$ scores.

We determine the significant edges by applying a best (optimal) thresholding technique presented in Appendix B (Algorithm 3) to binarize the learned adjacency matrices $A_D$ and $A_B$. Appendix B also provides sensitivity analysis of the best threshold (threshold–F1 score sensitivity curves) and a comparison between optimal and adaptive thresholding (Algorithm 4), supporting robust and fully data-driven binarization of both $A_D$ and $A_B$. When ground truth is absent, adaptive thresholding serves as a fully data-driven alternative.

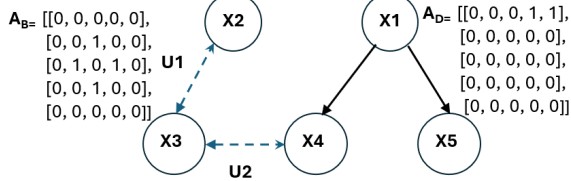
(a) Ground truth causal graph and adjacency matrices.

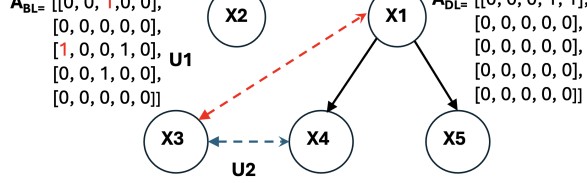
(b) Learned causal graph and adjacency matrices.

Figure 2: Comparison of ground-truth (left) and learned (right) causal graphs for FC. Solid: directed; dashed: bidirected; red: incorrect edges. Adjacency matrices: ground-truth: $A_D$, $A_B$; learned: $A_{DL}$, $A_{BL}$.

In case of **FC**, we train the model with 5000 epochs using ADAM optimizer with `ExponentialDecay` (Goodfellow et al., 2016) learning rate scheduling. We apply KL annealing with *kl_weight* (a gradually increasing weight) until epoch 50 to multiply the KL divergence term to counter KL-vanishing during the initial training phase and focus only on reconstruction error minimization. The causal transition epoch (CTE) is set to 150 to learn the data distribution and input characteristics well before applying the causal regularization. The full configuration details are available in the Appendix E (Table 9), along with their sensitivity analysis.

The learned causal graph and adjacency matrices are shown in Figure 2b. **All directed** cause-effect relations are correctly identified by the proposed method. For bidirected confounding, the model correctly detects the edge between $X_3$ and $X_4$, the more complex confounding pattern, since $X_4$ participates in multiple causal

dependencies, having both a directed and bidirected influence, but misses the ground-truth bidirected edge $X_3 \leftrightarrow X_2$ and instead introduces a spurious edge $X_3 \leftrightarrow X_1$, representing a close confounding approximation and demonstrating the model's ability to infer approximate latent structures even in challenging scenarios. A red edge signifies a relationship absent in the ground truth.

For **ER**, each dataset ER(4,6,4), and ER(12,50,10) we train the proposed model using the ADAM optimizer with appropriate learning rate scheduling (either `ExponentialDecay` or `CosineDecay`), KL annealing to gradually increase the *kl_weight* to 1.5, and dataset-specific regularization coefficients. The complete configuration details, including learning schedules, latent dimensions, and regularization parameters, are provided in the Appendix E (Table 9). Figure 3a shows the ground-truth ER(4,6,4) causal graph and Figure 3b

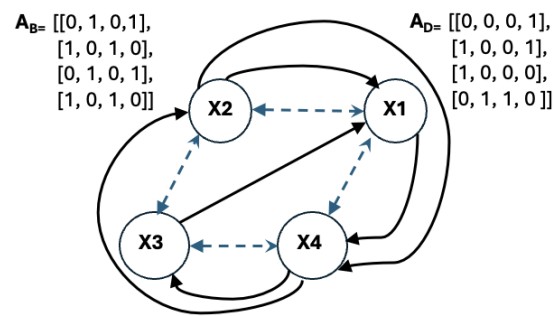

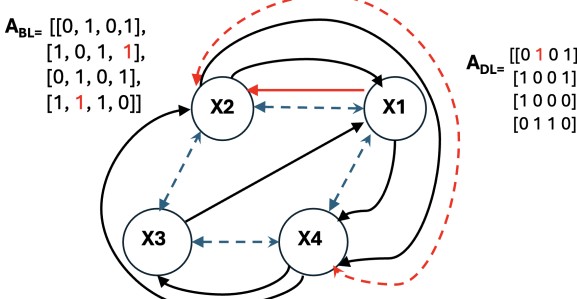

(a) Ground truth causal graph and adjacency matrices.       (b) Learned causal graph and adjacency matrices.

Figure 3: Comparison of ground-truth (left) and learned (right) causal graphs for ER(4,6,4). Solid: directed; dashed: bidirected; red: incorrect edges. Adjacency matrices: ground-truth: $A_D$, $A_B$; learned: $A_{DL}$, $A_{BL}$.

shows the corresponding learned structure. CTE plays an important role in the training process.

The comparative results are summarized in Table 1, demonstrating that our method delivers consistently competitive performance and, in certain scenarios, outperforms state-of-the-art approaches across multiple datasets. To assess robustness, we further evaluate our method on multiple independently sampled ER graphs (Appendix C), reporting mean ± std performance over five independent runs in Table 1. A detailed analysis of worst-case bidirected performance (including ER(12,50,10)), together with limitations and failure modes, is provided in Appendix G. For fairness and reproducibility, we retain the best-reported values for all baseline methods.

Table 1: F1 score comparison ($F1_D$: directed, $F1_B$: bidirected) across FC and ER

| Method | FC | | ER(4,6,4) | | ER(12, 50, 10) | |
|---|---|---|---|---|---|---|
| | $F1_D$ | $F1_B$ | $F1_D$ | $F1_B$ | $F1_D$ | $F1_B$ |
| FCI | 0.00 | 0.75 | 0.50 | 0.40 | 0.25 | 0.33 |
| CAM-UV | 0.80 | 0.67 | 0.30 | 0.25 | 0.38 | 0.36 |
| RCD | 0.00 | 0.54 | 0.35 | 0.35 | 0.45 | 0.20 |
| DCD | 0.00 | 0.67 | 0.25 | 0.20 | 0.32 | 0.18 |
| N-DAG-G | 0.50 | 0.00 | 0.60 | 0.00 | 0.55 | 0.00 |
| N-ADMG-G | 0.49 | **0.99** | 0.75 | 0.60 | 0.60 | 0.38 |
| N-BF-ADMG-G | 0.64 | 0.93 | 0.78 | 0.80 | **0.60** | 0.40 |
| **Proposed (G-ADMG-CL)** | **1.0 (0.00)** | 0.50 (0.00) | **0.84 (0.05)** | **0.89 (0.03)** | 0.53 (0.05) | **0.41 (0.05)** |

For the FC dataset, the proposed model achieves perfectly stable recovery with $F1_D = 1.0(0.00)$ and $F1_B = 0.50(0.00)$ across all runs. The bidirected score of 0.50 corresponds to a close confounding approximation ($X_3$ paired with $X_1$ instead of $X_2$), reflecting subtle latent-dependency ambiguity rather than model instability.

For the ER dataset, our method achieves the highest $F1_{\mathbf{D}}$ score of 0.92 and the highest $F1_{\mathbf{B}}$ score of 0.89 on the ER(4,6,4) graph (Figure 3), effectively recovering both directed and bidirected causal relationships. Averaged across multiple runs, the method attains an $F1_{\mathbf{D}}$ score of 0.84 and an $F1_{\mathbf{B}}$ score of 0.89 on ER(4,6,4). On the more complex ER(12,50,10) graph, the model attains a competitive average $F1_{\mathbf{D}}$ score of 0.53 (best: 0.58), and an $F1_{\mathbf{B}}$ score of 0.41 (best: 0.45), outperforming all baseline methods in detecting unobserved confounded relationships. These findings demonstrate the robustness of the proposed methodology in dense and high-dimensional confounding scenarios.

For the protein–signaling dataset (Sachs), Appendix F presents preliminary structure-estimation results showing the applicability of G-ADMG-CL in a complex biochemical system.

**Ablation study:** We perform ablation study for the causal relationships learning task.

$\mathcal{L}_{\mathbf{Causal\_ADMG}}$: The ablation study on causal mixed graph loss $\mathcal{L}_{\mathrm{Causal\_ADMG}}$, presents the significance of the proposed causal constraint applied to the loss function of G-ADMG-CL.

FC: We observe that removing causal constraint $\mathcal{L}_{\mathrm{Causal\_ADMG}}$ causes a significant reduction in the performance in FC. As a next step to establish the importance of the different components of causal mixed graph loss like bow-free constraint $\lambda_{\mathrm{bow}}$, symmetry/asymmetry constraints of adjacency matrices, we set their value to zero to remove their influence. We alter the value of $\lambda_{\mathrm{cycle}}$ to a high value to judge its sensitivity, as depicted in Table 2. This study establishes that all these components significantly impact performance and play a significant role in causal structure learning.

ER: We observe that removing $\mathcal{L}_{\mathrm{Causal\_ADMG}}$ causes a significant reduction in the performance as depicted in Table 2. As a next step to establish the importance of the different components of this causal mixed graph loss we perform the same steps as performed in FC. We notice that the symmetry constraint of $A_B$ plays an important role. ER data has overlapping directed and bidirected edges where bow-free constraint aims to ensure non overlapping directed and bidirected edges which reduces the impact of the bow effect. This study also reflects that our method performs well in the presence of overlapping directed and bidirected edges, that is, when variables have both a direct cause-effect relationship and are influenced by a common unobserved confounder.

The **additional ablation study** in Appendix E.2 shows that strong bow-free regularisation and moderate sparsity improve recovery of both directed and bidirected structures, while excessive regularisation can degrade performance.

Table 2: Ablation study of $\mathcal{L}_{\mathrm{Causal\_ADMG}}$ and its components using F1 score

| Method | FC | | ER(4,6,4) | |
|---|---|---|---|---|
| | $F1_{\mathbf{D}}$ | $F1_{\mathbf{B}}$ | $F1_{\mathbf{D}}$ | $F1_{\mathbf{B}}$ |
| Proposed with $\mathcal{L}_{\mathrm{total}}$ | 1.0 | 0.50 | 0.92 | 0.89 |
| $\mathcal{L}_{\mathrm{total}} - \mathcal{L}_{\mathrm{Causal\_ADMG}}$ | 0.50 | 0.31 | 0.86 | 0.78 |
| $\mathcal{L}_{\mathrm{Causal\_ADMG}} - \mathcal{L}_{\mathrm{bow}}$ | 0.44 | 0.5 | 0.86 | 0.88 |
| $\mathcal{L}_{\mathrm{Causal\_ADMG}} - \mathcal{L}_{\mathrm{symmetry}}(A_B)$ | 0.66 | 0.25 | 0.86 | 0.82 |
| $\mathcal{L}_{\mathrm{Causal\_ADMG}} - \mathcal{L}_{\mathrm{asymmetry}}(A_D)$ | 0.66 | 0.5 | 0.92 | 0.89 |
| $\mathcal{L}_{\mathrm{Causal\_ADMG}} : \lambda_{\mathrm{cycle}} = 10$ | 0.57 | 0.40 | 0.92 | 0.89 |

**Ablation study on causal annealing:** This study presents the sensitivity of the proposed causal annealing applied to the G-ADMG-CL, the causally constrained VAE. We compare the baseline G-ADMG-CL without annealing, ($\lambda_{\mathrm{causal}} = 0$) against G-ADMG-CL with hard causal annealing ($\lambda_{\mathrm{causal}} = 1$). Table 3 presents an ablation study to evaluate the impact of causal annealing.

FC: In this dataset, we observe that using baseline G-ADMG-CL reduces the performance for directed relationships on the FC dataset ($F1_{\mathrm{D}} = 0.50$) significantly. In contrast, G-ADMG-CL with annealing achieves the highest directed $F1_{\mathrm{D}} = 1.00$ performance.

Table 3: Ablation study of causal annealing using F1 score

| Method | FC | | ER(4,6,4) | |
|---|---|---|---|---|
| | $F1_{\mathbf{D}}$ | $F1_{\mathbf{B}}$ | $F1_{\mathbf{D}}$ | $F1_{\mathbf{B}}$ |
| **G-ADMG-CL** (with Causal Annealing) | **1.0** | 0.50 | **0.92** | **0.89** |
| G-ADMG-CL (without Causal Annealing) | 0.50 | 0.50 | 0.75 | 0.80 |

ER: For ER, we notice that using baseline G-ADMG-CL reduces the performance for directed relationships on the ER dataset ($F1_{\mathrm{D}} = 0.75$) significantly, and ($F1_{\mathrm{B}} = 0.80$). In contrast, G-ADMG-CL with annealing achieves the highest directed $F1_{\mathrm{D}}$ on ER (0.92) and maintains strong bidirected performance (0.89). All ablation experiments are conducted using the best-performing configuration.

In addition to employing hard causal annealing, early evaluations are conducted using a linear causal-annealing schedule on the ER(12,50,10) graph, as detailed in Appendix E.2.1. Overall, linear scheduling demonstrates comparable effectiveness to hard causal annealing.

### 5.3 Causal Inference

We demonstrate the applicability of our method to a **causal inference task** using the IHDP dataset. While we refer to the causal inference extension of our model as G-ADMG-CL+P, we report results and comparisons using the core model name **G-ADMG-CL**, consistent with baseline naming conventions. We first train the model using IHDP_train to obtain the learned directed and bidirected latent representations. The model is trained for 5000 epochs using the ADAM optimizer, with ExponentialDecay as the learning scheduler. The initial learning rate is set at 0.001, with *decay_steps*=1000, and *decay_rate* = 0.90. To address KL vanishing, we apply KL annealing, where the KL divergence term is multiplied by a gradually increasing *kl_weight* until it reaches 1.5 at epoch 20, allowing the model to focus on reconstruction error minimization during early training. The dimensionality of both latent spaces $z_{\mathrm{D}}$ and $z_{\mathrm{B}}$ is set to 50, as this data is relatively complex. We consider $\lambda_{\mathrm{cycle}} = 1$, $\lambda_{\mathrm{bow}} = 5$, $\lambda_{\mathrm{entropy}}(A_D) = 0.01$, $\lambda_{\mathrm{entropy}}(A_B) = 0.001$, $\lambda_{\mathrm{asymmetry}}(A_D) = 0.05$, $\lambda_{\mathrm{symmetry}}(A_B) = 4.75$. The causal transition epoch (CTE) is set to 1000 to ensure that the G-ADMG-CL learns the data distribution and input characteristics well before applying the causal regularization.

We apply the trained model, G-ADMG-CL, to the IHDP_test for obtaining the predicted outcome $y_{\mathrm{treated\_P}}$, $y_{\mathrm{control\_P}}$ as shown in Eq. (16), Eq. (17) using the decoder reconstructed output as described in the methodology.

$$model = \text{G-ADMG-CL}(\text{IHDP\_train}) \tag{16}$$

$$y_{\mathrm{treated\_P}}, y_{\mathrm{control\_P}} = model.predict(\text{IHDP\_test}) \tag{17}$$

Table 4 shows that the proposed method outperforms existing state-of-the-art methods by achieving the

Table 4: Causal inference results using the IHDP dataset

| Method | RMSE-ATE |
|---|---|
| FCI | 0.13 |
| CAM-UV | 0.15 |
| RCD | 0.14 |
| DCD | 0.16 |
| N-DAG-G | 0.12 |
| N-BF-ADMG-G | 0.10 |
| G-ADMG-CL | **0.031** |

lowest RMSE-ATE. Since the IHDP does not provide a ground-truth causal graph, the ATE serves as the most reliable metric for comparing causal structure learning performance.

For completeness, Appendix H provides a consolidated overview of the supplementary experimental analyses, including threshold selection, ER reproducibility, hyperparameter sensitivity, causal annealing behaviour, and early evaluation on the Sachs protein–signaling dataset.

## 6 Conclusion

We propose a novel VAE-based generative method for learning causal relationships under latent confounding. A key novelty of our architecture is the use of **dual latent spaces**, which separately encode directed cause–effect relationships and bidirected latent confounded influences an ability not present in existing VAE-based causal discovery methods. These latent representations associate with the trainable adjacency matrices $A_D$ and $A_B$, enabling ADMG-based structure learning. We introduce the causal mixed-graph loss $\mathcal{L}_{\text{Causal\_ADMG}}$ to enforce acyclicity and capture bidirected dependencies. Another contribution is the **causal annealing** mechanism, which progressively activates causal constraints during training and is absent in prior causal structure learning approaches. Ablation studies confirm the importance of both components. Our method, **G-ADMG-CL**, achieves competitive or improved performance over established baselines, and the learned structure improves causal inference on real data.

Future work includes exploring annealing schedules (CTE effects) and analyzing directed–bidirected edge coexistence in complex graphs. Another direction is modelling *complex and hierarchical confounders* to improve bidirected recovery under densely confounded graphs. We aim to extend the evaluation of our method to physics-based simulators such as **TriFinger** (Wüthrich et al., 2021) and **Meta-World** (Yu et al., 2019a), which exhibit rich nonlinear dynamics and multi-object dependencies, providing a challenging testbed for causal structure learning.

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

## Appendix

## Overview

This appendix provides supplementary analyses and experimental details that support and extend the findings presented in the main paper, organized as follows:

- **ER data synthesis (Appendix A):** This section describes the ER graph construction, sampling protocol, and the nonlinear SCM used for training and evaluation.

- **Threshold selection and sensitivity analysis (Appendix B):** This section first presents the thresholding algorithms used to binarize the learned adjacency matrices, followed by a sensitivity analysis showing that directed recovery is robust to threshold variation and that adaptive thresholding provides a fully data-driven alternative when ground truth is unavailable.

- **Additional results on ER (Appendix C):** This section reports reproducibility analysis for the ER(4,6,4) and ER(12,50,10) settings, presenting performance across multiple independently sampled graphs.

- **ADMG estimation and causality disclaimer (Appendix D):** This section clarifies the assumptions underlying ADMG estimation and the limitations of recovering true latent confounding structures.

- **Hyperparameter sensitivity and training configurations (Appendix E):** This section comprises the following: Appendix ( E.1) reports the hyperparameter settings used for the synthetic datasets. Appendix E.2 presents an additional ablation study on the regularisation coefficients, while Appendix E.2.1 analyses the effect of linear causal annealing, highlighting the impact of bow-free constraints, sparsity, and annealing schedules.

- **Additional experiment: Sachs protein–signaling dataset (early results) (Appendix F):** This section reports preliminary structure-estimation results on the Sachs dataset, illustrating the applicability of the method to real biochemical data with nonlinear dependencies.

- **Discussion on limitations (Appendix G):** This section analyses the scope and boundary conditions of the method.

- **Summary of experimental results (Appendix H):** This section provides a consolidated overview of the additional experimental studies reported in the Appendix, including threshold analyses, ER reproducibility experiments, hyperparameter ablations, causal annealing schedules, and real-world evaluation.

## A   ER Data Synthesis

**ER Configuration:**   The ER-based graphs are generated using three parameters: $d$, $e$, and $m$.

- $d$: the number of observed variables or nodes in the causal graph,

- $e$: the number of directed edges representing direct cause-effect relationships,

- $m$: the number of bidirected edges modeling unobserved confounding between variables.

For example, ER(4,6,4) denotes a 4-variable graph with 6 directed edges and 4 bidirected edges. Similarly, ER(12,50,10) corresponds to a larger graph with 12 variables, 50 directed edges, and 10 bidirected edges.

These settings allow us to evaluate how the model performs under increasing structural complexity and the presence of latent confounders. We use the Eq. 18 to generate ER data.

$$A_D \sim \text{ER}\left(d, \frac{e}{d(d-1)}\right), \quad \text{diag}(A_D) = 0$$

$$A_D[i,j] = \begin{cases} 1 & \text{if there is a directed edge from } i \text{ to } j, \\ 0 & \text{otherwise.} \end{cases}$$

$$A_B[i,j] \sim \text{Bernoulli}\left(\frac{m}{d(d-1)}\right), \quad \text{diag}(A_B) = 0$$

$$A_B = \text{triu}(A_B, 1) + \text{triu}(A_B, 1)^\top,$$

$$: \text{triu extracts the elements above the diagonal,}$$

$$\epsilon \sim \mathcal{N}(0, 0.1^2), \quad u \sim \mathcal{N}(0, 0.1^2),$$

$$X_i = \sum_{p \in \text{Pa}_D(i)} f(X_p) + \sum_{q \in \text{Pa}_B(i)} g(u_q) + \epsilon_i. \tag{18}$$

## B    Threshold Selection and Sensitivity Analysis

In this section, we present the threshold selection method. We also describe the procedure for identifying the optimal threshold to obtain the best F1 score for the learned adjacency matrices.

### B.1    Algorithm for Optimal Threshold Selection

To obtain the best-directed ($F1_\text{D}$) and bidirected ($F1_\text{B}$) adjacency F1 scores, we employ the method described in Algorithm 3. Figure 4 illustrates the variation of F1 scores with respect to the threshold values to obtain

---

**Algorithm 3** Optimal Threshold Selection for F1 score

---

1: **Input:** Ground truth adjacency matrix $A$, learned adjacency matrix $W$, set of thresholds $\mathcal{T}$
2: **Output:** Best threshold $t^*$, maximum F1 score $F1_\text{max}$
3: $F1_\text{max} \leftarrow 0, \quad t^* \leftarrow 0$
4: **for** each threshold $t \in \mathcal{T}$ **do**
5: $\quad W_\text{bin} \leftarrow \mathbb{I}(|W| \geq t)$          ▷ Binarize $W$ at threshold $t$
6: $\quad \mathbf{a} \leftarrow \text{flatten}(A)$          ▷ Convert matrix $A$ to vector
7: $\quad \mathbf{w} \leftarrow \text{flatten}(W_\text{bin})$          ▷ Convert $W_\text{bin}$ to vector
8: $\quad F1_t \leftarrow \text{F1\_score}(\mathbf{a}, \mathbf{w})$
9: $\quad$ **if** $F1_t > F1_\text{max}$ **then**
10: $\quad\quad F1_\text{max} \leftarrow F1_t$
11: $\quad\quad t^* \leftarrow t$
12: $\quad$ **end if**
13: **end for**
14: **return** $t^*, F1_\text{max}$

---

the optimal threshold selection for ER(12, 50, 10). Figure 4a shows the best-obtained F1 score for the directed adjacency matrix $A_D$, while Figure 4b depicts the best-obtained F1 score for the bidirected adjacency matrix $A_B$. The optimal (best) threshold, corresponding to the peak F1 value in each curve, is selected as described in Algorithm 3. The optimal threshold selection plots for the directed and bidirected adjacency matrices on the FC dataset are shown in Figure 5, highlighting the threshold values that yield the highest F1 score for each edge type.

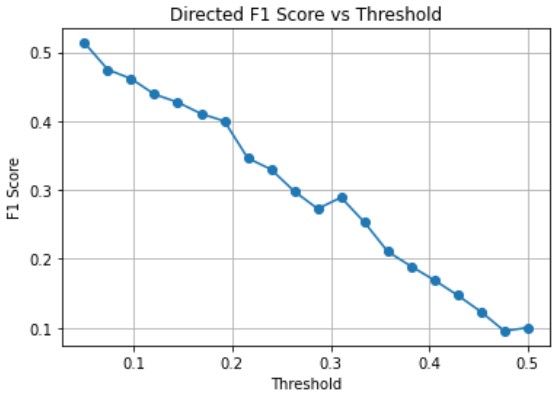
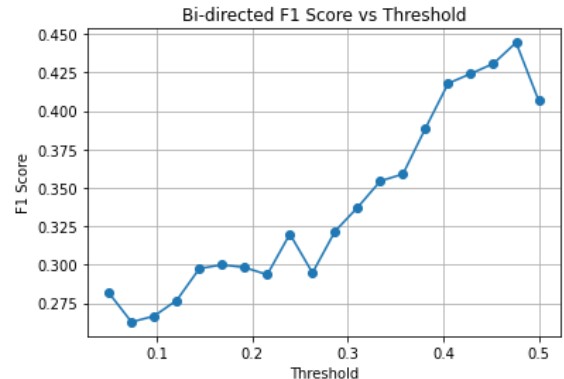

(a) Obtained best F1 score for learned $A_D$ .     (b) Obtained best F1 score for learned $A_B$.

Figure 4: Optimal threshold selection for ER(12, 50, 10).

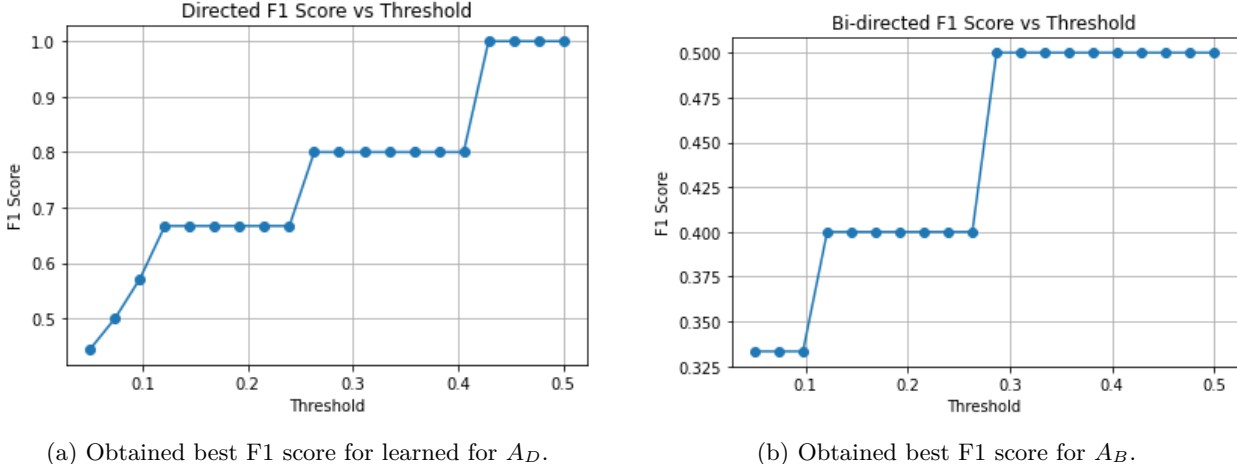

(a) Obtained best F1 score for learned for $A_D$.     (b) Obtained best F1 score for $A_B$.

Figure 5: Optimal threshold selection for FC.

**Sensitivity analysis of thresholding:** This sensitivity analysis is conducted on a random ER graph ER(12, 50, 10). Once trained, the threshold parameter is varied to analyze F1 score stability.

The model is evaluated across multiple threshold ranges, as shown in Table 5. Reported are the best-directed ($A_D$) and bidirected ($A_B$) F1 scores with their corresponding optimal thresholds ($\tau_D$, $\tau_B$). The final row shows mean $\pm$ standard deviation across all settings.

In this example, across all threshold ranges, the directed-edge ($A_D$) F1 scores exhibit a mean of $0.517\pm0.006$, confirming stable directed structure learning with only minor numerical fluctuation. Although their standard deviation is slightly higher than that of the bidirected edges, the corresponding optimal thresholds ($\tau_D = 0.04 \pm 0.01$) remain narrowly concentrated, demonstrating strong robustness to threshold perturbation.

For the bidirected edges ($A_B$), the mean F1 score of $0.346 \pm 0.005$ shows slightly lower numerical variation but a wider threshold spread ($\tau_B = 0.34 \pm 0.11$), reflecting moderate sensitivity due to latent confounding.

Overall, these results confirm that the learned adjacency matrices remain quantitatively stable under threshold variation, with directed relations showing consistent robustness and bidirected ones exhibiting moderate sensitivity linked to confounding effects.

In Figure 6 each pair of subplots shows the variation of F1 scores with changing binarization thresholds for (**top**) directed edges ($A_D$) and (**bottom**) bidirected edges ($A_B$). Directed F1 score remains consistently

Table 5: Threshold sensitivity analysis on ER(12, 50, 10).

| Count | Threshold Range | Best F1$_D$ | $\tau_D$ | Best F1$_B$ | $\tau_B$ |
|---|---|---|---|---|---|
| 1 | 0.01–0.50 | 0.507 | 0.05 | 0.341 | 0.33 |
| 2 | 0.02–0.50 | 0.519 | 0.02 | 0.348 | 0.57 |
| 3 | 0.03–0.50 | 0.519 | 0.03 | 0.352 | 0.25 |
| 4 | 0.04–0.50 | 0.519 | 0.04 | 0.341 | 0.33 |
| 5 | 0.01–0.60 | 0.522 | 0.04 | 0.348 | 0.24 |
| **Mean ± SD** | — | **0.517 ± 0.006** | **0.04 ± 0.01** | **0.346 ± 0.005** | **0.34 ± 0.11** |

(a) 1: $A_D$    (b) 2: $A_D$    (c) 3: $A_D$    (d) 4: $A_D$    (e) 5: $A_D$

(f) 1: $A_B$    (g) 2: $A_B$    (h) 3: $A_B$    (i) 4: $A_B$    (j) 5: $A_B$

Figure 6: Optimal threshold selection for threshold sensitivity analysis on ER(12, 50, 10).

around 0.5 across small threshold shifts (0.02–0.05), while bidirected F1 score peaks near 0.3 for thresholds between 0.25–0.33, confirming stable causal discovery under latent confounding. In all experiments, the best threshold is selected by varying over a fixed grid of candidate values shared across datasets, without any manual per-dataset tuning.

## B.2 Adaptive Thresholding

In realistic scenarios where the ground-truth causal graph is unavailable, the adaptive threshold mechanism (Algorithm 4) offers a principled alternative for threshold selection. However, since state-of-the-art methods typically report their best-obtained performance, we also compare against their best-achieved results to ensure fairness. As shown in Table 6, adaptive thresholding yields consistent F1 scores across runs, that are broadly consistent with best-threshold selection. The directed structure ($A_D$) demonstrates high stability (mean F1$_D$ = 0.508 vs. 0.536), while the bidirected component ($A_B$) shows moderate variability due to sensitivity to latent confounding density.

Table 6: Comparison of adaptive vs. best threshold F1 scores for directed ($A_D$) and bidirected ($A_B$) for different ER(12,50,10) graphs.

| Run | Adaptive F1$_D$ | Adaptive F1$_B$ | Best F1$_D$ | Threshold ($\tau_D$) | Best F1$_B$ | Threshold ($\tau_B$) |
|---|---|---|---|---|---|---|
| 1 | 0.508 | 0.304 | 0.516 | 0.12 | 0.333 | 0.29 |
| 2 | 0.492 | 0.348 | 0.507 | 0.05 | 0.341 | 0.33 |
| 3 | 0.525 | 0.304 | 0.586 | 0.05 | 0.308 | 0.26 |
| **Mean** | **0.508** | **0.319** | **0.536** | **0.07** | **0.327** | **0.29** |

---

**Algorithm 4** Adaptive Thresholding for Unsupervised Binarization

---

1: **Input:** Learned adjacency matrix $W$, spread threshold $\eta$, scaling factor $\tau$
2: **Output:** Binary adjacency matrix $W_{\text{bin}}$
3: $W_{\text{abs}} \leftarrow |W|$                           ▷ Take element-wise absolute value
4: $w_{\max} \leftarrow \max(W_{\text{abs}}), \quad w_{\text{med}} \leftarrow \text{median}(W_{\text{abs}})$
5: $s \leftarrow \dfrac{w_{\max}}{(w_{\text{med}} + 10^{-8})}$                  ▷ Compute spread ratio
6: **if** $s \leq \eta$ **then**
7:      $\theta \leftarrow w_{\text{med}}$
8: **else**
9:      $\theta \leftarrow \tau \cdot w_{\max}$
10: **end if**
11: $W_{\text{bin}} \leftarrow \mathbb{I}(|W| \geq \theta)$                     ▷ Binarize adjacency matrix
12: **return** $W_{\text{bin}}$

---

**Reliability verification on thresholding methods:** To ensure robustness of the proposed structure learning, we further performed reliability testing on the FC dataset, which has a fixed causal graph. We compared three thresholding strategies: absolute value cutoff ($|W|_{\max} \geq 0.45$), adaptive thresholding, and best threshold search and observed consistent results across all settings. This confirms that the learned adjacency patterns remain stable under different thresholding criteria and that the proposed G-ADMG-CL framework yields reproducible causal structures across multiple evaluation protocols.

## C  Additional Results on ER

In this section we provide reproducibility analysis for the ER(4, 6, 4) and ER(12, 50, 10) configurations. Each run corresponds to a new random graph sampled under the respective ER settings. Table 7 and Table 8 report F1 scores across five such runs. Notably, the bidirected recovery remains relatively stable across

Table 7: Additional results for ER(4,6,4) dataset with five independent graphs.

| Run | $F1_{\text{D}}$ | $F1_{\text{B}}$ |
|---|---|---|
| 1 | 0.800 | 0.933 |
| 2 | 0.800 | 0.842 |
| 3 | 0.857 | 0.880 |
| 4 | 0.800 | 0.889 |
| 5 | 0.920 | 0.889 |
| **Mean $\pm$ Std** | **0.835 $\pm$ 0.051** | **0.887 $\pm$ 0.031** |

Table 8: Additional results for ER(12, 50, 10) with five independent graphs.

| Run | **F1$_{\text{D}}$** | **F1$_{\text{B}}$** |
|---|---|---|
| 1 | 0.576 | 0.316 |
| 2 | 0.589 | 0.450 |
| 3 | 0.538 | 0.414 |
| 4 | 0.510 | 0.450 |
| 5 | 0.452 | 0.424 |
| **Mean $\pm$ Std** | **0.533 $\pm$ 0.049** | **0.411 $\pm$ 0.052** |

runs, highlighting the robustness of our model in consistently capturing latent confounding even in dense, high-dimensional graphs such as ER(12,50,10).

## D  ADMG Estimation and Causality Disclaimer

ADMGs provide a principled framework for representing causal structure under latent confounding; however, learning a causal graph from observational data alone is fundamentally limited (Peters et al., 2017; Pearl & Bareinboim, 2014). Continuous-optimization approaches cannot guarantee recovery of the exact ground-truth SCM, since multiple ADMGs may induce the same observational distribution (Markov equivalence) (Spirtes et al., 2000), and the placement of latent confounders may remain ambiguous.

In alignment with prior work (Ashman et al., 2023), we follow ADMG structure estimation assumptions that allow for latent confounding, offering identifiability criteria for ADMGs in nonlinear additive-noise SCMs, although it does not ensure the structural identifiability of the underlying magnified SCM. Within this framework, each variable is represented as a nonlinear function of its direct causes, combined with independent noise and exogenous latent confounders, with no interaction effects between the observed and latent.

While these assumptions provide theoretical grounding, practical recoverability may still be limited when real data violate the additive-noise or independence conditions. Accordingly, the proposed method learns one ADMG structure that is consistent with the observed data; however, it does *not* constitute definitive causal proof, rather, the learned graph should be interpreted as a plausible causal hypothesis obtained through robust ADMG *estimation*, supported by the data and the adopted modelling assumptions.

## E  Hyperparameter Sensitivity and Training Configurations

### E.1  Training Configurations for Synthetic Datasets

We present the training configuration table for the synthetic data FC and ER below Table 9.

Table 9: Training configurations for FC, ER(4,6,4), and ER(12,50,10) experiments. Hyperparameters include KL annealing, learning rate schedules, and regularization weights ($\lambda$) for entropy, symmetry, cycle, and bow-free constraints.

| Setting | FC | ER(4,6,4) | ER(12, 50, 10) |
|---|---|---|---|
| Epochs | 5000 | 3000 | 2000 |
| Optimizer | ADAM | ADAM | ADAM |
| Learning Rate Schedule | ExponentialDecay | CosineDecay | ExponentialDecay |
| Initial Learning Rate | 0.001 | 0.001 | 0.001 |
| Decay Steps | 1000 | 1000 | 64 |
| Decay Rate | 0.90 | – | 0.98 |
| Minimum Learning Rate ($\alpha$) | – | $1 \times 10^{-5}$ | – |
| KL Annealing End Epoch | 50 | 100 | 800 |
| Final *kl_weight* | 1.5 | 1.5 | 1.5 |
| Latent Dimension | $z_{\text{dir}}$: 24, $z_{\text{bidir}}$: 24 | 24 | 36 |
| $\lambda_{\text{cycle}}$ | 1 | 7 | 5 |
| $\lambda_{\text{bow}}$ | 5 | 5 | 5 |
| $\lambda_{\text{entropy}}(A_D)$ | 0.005 | 0.001 | 0.001 |
| $\lambda_{\text{entropy}}(A_B)$ | 0.01 | 0.01 | 0.001 |
| $\lambda_{\text{asymmetry}}(A_D)$ | 0.05 | 0.05 | 0.05 |
| $\lambda_{\text{symmetry}}(A_B)$ | 0.5 | 1.5 | 1.75 |
| $\lambda_{\text{sparsity}}(A_D)$ | 0.001 | 0.001 | 0.001 |
| $\lambda_{\text{sparsity}}(A_B)$ | 0.02 | 0.001 | 0.001 |
| Causal Transition Epoch (CTE) | 150 | 150 | 1000 |

### E.2 Additional Ablation Study

$\lambda_{\textbf{bow}}$ **sensitivity:**   We first tune $\lambda_{\text{bow}}$ to enforce bow-freeness.

We performed an ablation over the bow-free coefficient $\lambda_{\text{bow}}$ on the FC data, varying it in $[1.0, 5.0]$, depicted in Figure 7. The $\text{F1}_D$ increased with $\lambda_{\text{bow}}$ and peaked at $\lambda_{\text{bow}} \in \{4.0, 5.0\}$ ($\text{F1}_D = 1.0$), confirming that a sufficiently strong bow-free constraint is beneficial. We therefore fix $\lambda_{\text{bow}} = 5.0$ in all main experiments and report the sensitivity curve.

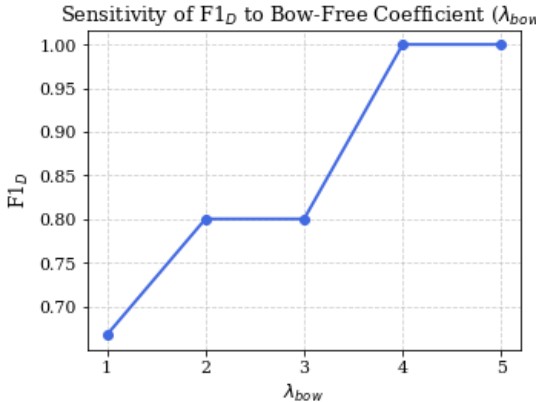

Figure 7: Sensitivity of $\text{F1}_D$ to the bow-free regularization coefficient ($\lambda_{bow}$) on the FC dataset. The model achieves the highest $\text{F1}_D$ at $\lambda_{bow} \in \{4, 5\}$, confirming that moderate penalization yields optimal separation between directed and bidirected dependencies.

**Sparsity sensitivity:**   We further analyze the effect of the sparsity coefficient $\lambda_{\text{sparse}}$ on the bidirected weight matrix $\mathbf{W_2}$ using the FC dataset. As shown in Table 10, the directed-edge accuracy ($\text{F1}_D$) remains consistently perfect across all values, while the bidirected $\text{F1}_B$ improves slightly with higher sparsity, peaking at $\lambda_{\text{sparse}} \in \{0.02, 0.03\}$ ($\text{F1}_B = 0.5$). Higher sparsity (0.04) begins to prune true directed edges. This indicates that moderate sparsity encourages better disentanglement of confounded relations without degrading directed structure recovery.

Table 10: Effect of sparsity coefficient on structure recovery (FC dataset).

| $\lambda_{\text{sparse}}$ | $\text{F1}_D$ | $\text{F1}_B$ |
|---|---|---|
| 0.005 | 1.0 | 0.4 |
| 0.01 | 1.0 | 0.4 |
| 0.02 | 1.0 | 0.5 |
| 0.03 | 1.0 | 0.5 |
| 0.04 | 0.8 | 0.5 |

**Guidelines for setting structural hyperparameters:**   In practice, we tune $\lambda_{\text{bow}}$ first to ensure bow-freeness, then adjust $\lambda_{\text{sparse}}$ and $\lambda_{\text{entropy}}$. For datasets with *structured and sparse* confounding (e.g., FC), a moderate increase in $\lambda_{\text{sparse}}^{(B)}$ helps suppress weak spurious links while keeping $\text{F1}_D$ stable. Excessively large $\lambda_{\text{cycle}}$ should be avoided, as it can remove genuine directed dependencies.

For the directed adjacency $\mathbf{A_D}$, we keep the *entropy* weight low to avoid randomization of edges, and use sparsity with a balanced ratio between sparsity and entropy:

$$\rho_D \triangleq \frac{\lambda_{\text{sparse}}^{(D)}}{\lambda_{\text{entropy}}^{(D)}}$$

This keeps $\mathbf{A_D}$ parsimonious without injecting noise; excessively high sparsity or entropy can hamper the recovery of true directed relations.

For the bidirected adjacency $\mathbf{A_B}$, the choice of sparsity depends on the nature of the latent confounding. When confounding is *sparse and structured* (e.g., FC), a higher $\lambda_{\text{sparse}}^{(B)}$ improves disentanglement of true bidirected edges. However, for dense and randomly distributed confounding patterns (as in ER(12,50,10)), a *smaller sparsity weight* is preferred, because strong sparsity prunes true confounding edges and destabilizes symmetry, leading to degraded F1 performance. Thus, sparsity must be tuned per dataset rather than using a global value.

**Symmetry–asymmetry balance:** To preserve the structural properties of ADMGs, the symmetry regularization on $\mathbf{A_B}$ must dominate the asymmetry penalty on $\mathbf{A_D}$:

$$\lambda_{\text{symmetry}}^{(B)} > \lambda_{\text{asymmetry}}^{(D)}.$$

This enforces a clean separation between directed and bidirected components, preventing leakage of asymmetric patterns into $\mathbf{A_B}$ and avoiding spurious symmetric patterns in $\mathbf{A_D}$. In practice, we choose $\lambda_{\text{symmetry}}^{(B)} \in [1.0, 5.0]$ and $\lambda_{\text{asymmetry}}^{(D)} \in [0.05, 0.1]$, maintaining a stable ratio that empirically improves identifiability.

In practice, we first tune $\lambda_{\text{bow}}$ to enforce bow-freeness, then choose $(\lambda_{\text{sparse}}^{(D)}, \lambda_{\text{entropy}}^{(D)})$ with $\rho_D \approx 1$ and a small absolute entropy weight, and finally adjust $(\lambda_{\text{sparse}}^{(B)}, \lambda_{\text{entropy}}^{(B)})$ to control the density of $\mathbf{A_B}$ when graphs are potentially complex.

A more exhaustive sparsity–sensitivity analysis across a wider range of graph families and real-world datasets is beneficial, and we identify this as an important direction for future work.

**Over-sparsification effect:** While moderate sparsity ($\lambda_{\text{sparse}} \in [0.01, 0.03]$) preserves perfect directed recovery on FC (F1$_D = 1.0$) and slightly improves F1$_B$, a higher value ($\lambda_{\text{sparse}} = 0.04$) reduces F1$_D$ to 0.8, indicating over–sparsification. This confirms a trade-off: excessive sparsity can prune true directed edges. Accordingly, $\lambda_{\text{sparse}}^{(D)} \leq 0.03$ and recommend separate sparsity terms for directed and bidirected components, tuning $\lambda_{\text{sparse}}^{(B)}$ independently when confounding is complex.

**Practical observation during training:** In real-world scenarios, the true causal structure is unknown, and direct F1-based evaluation is infeasible. In such cases, monitoring the evolution of the learned adjacency matrices $\mathbf{A_D}$ and $\mathbf{A_B}$ during training provides useful qualitative guidance. Stable directed adjacency patterns with low entropy indicate convergence toward interpretable causal relations, whereas excessive sparsity or rapidly fluctuating weights suggest over-regularization or noisy learning. Hence, users can rely on the relative stability and sparsity of $\mathbf{A_D}$ and $\mathbf{A_B}$ as proxies to tune $\lambda_{\text{sparse}}$ and $\lambda_{\text{entropy}}$ when ground truth graphs are unavailable.

### E.2.1 Linear Causal Annealing (Initial Study on ER(12,50,10))

In addition to hard causal annealing (used in all main experiments), we conducted preliminary tests with a *linear causal–annealing schedule* on the more complex ER(12,50,10) graph, where the causal-regularization weight is kept at 0 for the first 500 epochs, then increased linearly over 1000 epochs, and finally fixed at 1. Across two independent batches of experiments linear causal annealing yields directed-edge F1 values in the range 0.45–0.57 and bidirected-edge F1 in the range 0.26–0.40. These values are comparable to those obtained under hard annealing, indicating that the linear schedule does not provide a systematic performance benefit under the current configuration.

Overall, the linear schedule yields performance **very similar** to that of hard causal annealing, with no significant or systematic improvement. This suggests that the proposed hard annealing scheme is already sufficiently stable for structure learning in practice. However a more exhaustive exploration of nonlinear annealing schedules (e.g., cosine or exponential warm-up) is a promising direction for future work but remains outside the scope of the current revision.

## F  Additional Experiment: Sachs Protein-Signaling Dataset (Early Results)

We further conducted preliminary experiments on the well-known **Sachs protein-signaling dataset** (Sachs et al., 2005), which models biochemical interactions among 11 observed variables (*PKC, P38, Jnk, PKA, Raf, Mek, Erk, Plcg, PIP2, PIP3, Akt*). The original dataset does not explicitly include latent confounders, but in practice, unmeasured experimental factors may introduce confounding effects. The dataset remains a widely used real-world benchmark for real-world causal structure discovery.

Sachs provides a curated directed graph but *does not define ground-truth bidirected edges*. Following prior causal-discovery practice, we construct a minimal heuristic reference set capturing known co-regulation patterns purely for qualitative assessment. Thus, $F1_B$ should be interpreted as an indicator of latent-dependency capture, not as strict accuracy.

**Ground-truth edges:**  The directed edges used for evaluation are:

$$PKC \rightarrow \{P38, \text{ Jnk}, \text{ Raf}, \text{ Mek}\}, \quad PKA \rightarrow \{Jnk, \text{ Raf}, \text{ Mek}, \text{ Erk}, \text{ Akt}\},$$
$$Raf \rightarrow Mek, \quad Mek \rightarrow Erk, \quad Plcg \rightarrow PIP2, \quad PIP2 \rightarrow PIP3, \quad PIP3 \rightarrow Akt.$$

Heuristic shallow bidirected reference edges (used only for qualitative comparison):

$$\{PKC \leftrightarrow Jnk, \text{ PKC} \leftrightarrow PIP3, \text{ Raf} \leftrightarrow Erk, \text{ Mek} \leftrightarrow Akt\}.$$

As an early evaluation result on the Sachs dataset, the proposed G-ADMG-CL model achieves a directed-edge

Table 11: Recovered directed and bidirected edges on the protein–signalling network using G-ADMG-CL (evaluated against the ground-truth adjacency $A_D$ used in our implementation).

| Edge Type | Recovered Edge |
|---|---|
| **Directed Edges** | |
| Correct | PKC $\rightarrow$ Jnk |
| Correct | PKA $\rightarrow$ Mek |
| Correct | PKA $\rightarrow$ Erk |
| Correct | PKA $\rightarrow$ Akt |
| Correct | PIP3 $\rightarrow$ Akt |
| Spurious | Jnk $\rightarrow$ Erk |
| Spurious | Mek $\rightarrow$ PIP3 |
| Spurious | Plcg $\rightarrow$ Raf |
| Spurious | PIP3 $\rightarrow$ P38 |
| Spurious | Mek $\rightarrow$ PIP3 |
| Spurious | Akt $\rightarrow$ Plcg |
| **Bidirected Edges** | |
| Correct (heuristic) | Raf $\leftrightarrow$ Erk |
| Spurious | P38 $\leftrightarrow$ Jnk |
| Spurious | PIP2 $\leftrightarrow$ P38 |
| Spurious | PKC $\leftrightarrow$ Akt |
| Spurious | PKC $\leftrightarrow$ Plcg |
| Spurious | Mek $\leftrightarrow$ Plcg |
| Spurious | Multiple weak cross-links |

recovery of $F1_D = 0.387$ at threshold 0.41 and a bidirected-edge recovery of $F1_B = 0.117$ at threshold 0.05 after training for 150 epochs with CTE = 50. For these experiments, we set the structural regularisation weights to $\lambda_{\text{cycle}} = 1$ and $\lambda_{\text{bow}} = 5$. A stronger sparsity weight is applied to the bidirected component ($\lambda_{\text{sparsity}}(A_B) = 0.1$) because increased sparsification helps suppress spurious bidirected links while preserving the directed structure, whereas a lower weight ($\lambda_{\text{sparsity}}(A_D) = 0.001$) is used for the directed component, following the hyperparameter guidelines in Appendix E. To assess binarisation on real data. Figure 8 presents

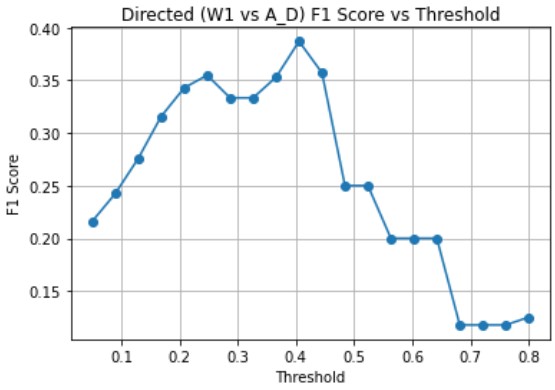

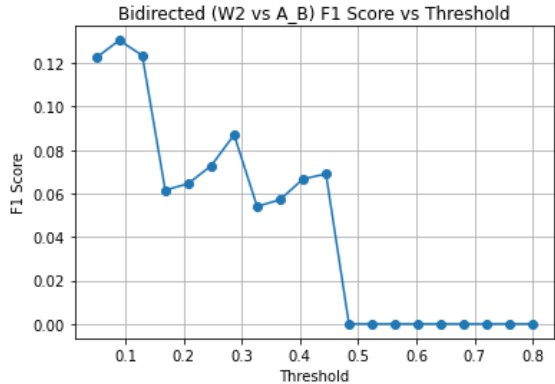

(a) Directed ($W_1$ vs $A_D$) F1 score vs threshold.

(b) Bidirected ($W_2$ vs $A_B$) F1 score vs threshold.

Figure 8: Optimal threshold analysis on the Sachs protein–signaling dataset

the optimal threshold curves for the Sachs dataset, and Table 11 reports the corresponding structure–recovery metrics.

These preliminary findings suggest that the proposed **G-ADMG-CL** framework is generalizable to real-world biochemical systems.

## G  Discussion on Limitations

**Shallow vs. complex confounded associations:**  The model reliably captures **structured** shallow confounding, including nodes that simultaneously participate in both *directed causal* and *latent confounded* relations, exemplified by the FC variable $x_4$. Recovering such mixed-role variables demonstrates the capacity of G-ADMG-CL to capture observed and unobserved causal relations within a unified ADMG framework. The method further maintains **robust** directed recovery in dense, high-dimensional settings, as evidenced by the ER(12,50,10) graph where it achieves an average $F1_D = 0.53$, outperforming existing ADMG learners under comparable conditions. Performance degradation may arise under extremely dense overlapping confounding where many variables share entangled latent causes, while directed edges remain stable, indicating **graceful degradation** of $A_B$ estimation.

**Case study on ER(12,50,10):**  To examine latent confounding recovery in detail, we analyze one representative ER(12,50,10) graph, where the model achieves $F1_D = 0.584$ and $F1_B = 0.308$. **Shallow confounding** denotes pairwise hidden causes (e.g., $X_2 \leftrightarrow X_5$), whereas **complex confounding** arises when multiple variables share overlapping latent dependencies. As shown in Table 12, the proposed method recovers most shallow confounding relations (e.g., $X_1 \leftrightarrow X_9$, $X_5 \leftrightarrow X_6$). In multi-variable settings (e.g., among $X_5$, $X_{10}$, $X_{11}$), additional weak bidirected links appear, explaining the moderate $F1_B = 0.308$ despite high recall.

**Worst-case bidirected performance:**  Across multiple independently sampled ER graphs, when confounding signals are weak, diffuse, or overlap with directed edges, bidirected recovery enters a worst-case regime. The ER(12,50,10) setting ($F1_B = 0.308$) is representative of this behaviour, where densely entangled latent structures make it difficult to distinguish true shared confounders from residual dependence. Directed-edge recovery remains **stable**, while bidirected estimation degrades gradually, providing an empirical lower bound on the ability to recover complex latent confounding from observational data alone.

**Limitations and failure modes:**  As with other continuous optimization-based causal discovery approaches, such as NOTEARS and DAG-GNN, our framework does not ensure unique recovery of the true underlying structural causal model across all possible data-generating processes. Rather, its objective is to achieve precise estimation of ADMGs.

Table 12: Comparison of ground truth vs. learned confounding relations at $\tau_B = 0.26$

| Pair ($\mathbf{X}_i \leftrightarrow \mathbf{X}_j$) | GT | Learned | Match |
|---|---|---|---|
| $X_1 \leftrightarrow X_9$ | 1 | 1 | ✓ |
| $X_1 \leftrightarrow X_{12}$ | 1 | 1 | ✓ |
| $X_5 \leftrightarrow X_6$ | 1 | 1 | ✓ |
| $X_5 \leftrightarrow X_7$ | 1 | 1 | ✓ |
| $X_5 \leftrightarrow X_{10}$ | 1 | 0 | ✕ |
| $X_6 \leftrightarrow X_2$ | 1 | 1 | ✓ |
| $X_6 \leftrightarrow X_3$ | 1 | 1 | ✓ |
| $X_7 \leftrightarrow X_5$ | 1 | 1 | ✓ |
| $X_7 \leftrightarrow X_8$ | 1 | 1 | ✓ |
| $X_7 \leftrightarrow X_{12}$ | 1 | 1 | ✓ |
| $X_9 \leftrightarrow X_1$ | 1 | 1 | ✓ |
| $X_{10} \leftrightarrow X_5$ | 1 | 0 | ✕ |
| $X_{10} \leftrightarrow X_{11}$ | 1 | 1 | ✓ |
| $X_{11} \leftrightarrow X_{10}$ | 1 | 1 | ✓ |
| $X_{12} \leftrightarrow X_1$ | 1 | 1 | ✓ |
| $X_{12} \leftrightarrow X_7$ | 1 | 1 | ✓ |
| **Total Correct** | 16 | 14 | **87.5%** |

As outlined below following practical failure modes may affect the recoverability of both the directed adjacency matrix $A_D$ and the bidirected adjacency matrix $A_B$.

1. **Weak causal effects:** When directed causal effects are small relative to noise, the induced statistical dependencies become insufficient for reliable detection, reducing the recoverability of edges in $A_D$.

2. **Near-deterministic relationships:** Extremely strong or near-deterministic functional relationships can collapse latent variability, leading to edge misclassification or spurious bidirected connections in $A_B$.

3. **Overlapping latent confounders:** When multiple latent confounders induce similar covariance patterns, the model may attribute confounding to incorrect variable pairs. In such cases, true bidirected edges in $A_B$ may be misinterpreted as weak directed edges in $A_D$, producing ambiguity between the two edge types.

4. **Highly correlated parents:** If several parents of a child variable exhibit strong correlation, the resulting dependence patterns may be statistically indistinguishable without interventions, creating unavoidable ambiguity in both $A_D$ and $A_B$.

These limitations reflect inherent challenges of observational causal discovery.

## H  Summary of Experimental Results

In addition to the robustness demonstrated in the main paper, this section provides a consolidated summary of the experimental studies and analyses reported in the Appendix.

- **Threshold sensitivity and adaptive thresholding (Appendix B):** We conduct sensitivity analysis on optimal (best) threshold selection to analyze F1 score stability. Directed structure recovery is robust to threshold variation, whereas bidirected recovery is more sensitive. Adaptive thresholding provides a fully data-driven alternative that performs comparably to best-threshold selection when ground truth is unavailable. We further verify threshold reliability using absolute-value cutoff, adaptive thresholding, and best-threshold selection, observing consistent results across all settings on the FC dataset.

- **Additional results on ER graphs (Appendix C):** Reproducibility analysis for the ER settings is described in this section, illustrating the consistent capture of latent confounding even in dense, high-dimensional settings, such as ER(12,50,10).

- **Additional ablation study (Appendix E.2):** We conduct an **additional ablation study** on the regularisation coefficients to analyse hyperparameter sensitivity. Increasing the bow-free regularisation coefficient $\lambda_{\text{bow}}$ improves directed edge recovery, with optimal results at $\lambda_{\text{bow}} \in \{4.0, 5.0\}$; $\lambda_{\text{bow}} = 5.0$ is used in subsequent experiments. For sparsity ($\lambda_{\text{sparse}}$), moderate values $\in \{0.02, 0.03\}$ enhance bidirected F1 without harming directed accuracy, while higher sparsity (0.04) removes true edges, revealing a trade-off between interpretability and robustness. Separate sparsity terms for directed and bidirected matrices are recommended, particularly under complex confounding. Practical guidelines include tuning $\lambda_{\text{bow}}$ first, balancing sparsity and entropy for directed edges, and setting bidirected sparsity according to dataset-specific confounding. Symmetry regularisation should dominate asymmetry penalisation to ensure clear separation of dependency types. Monitoring adjacency matrix evolution during training provides useful diagnostics when ground truth is unavailable.

  Overall, strong bow-free regularisation and moderate sparsity support accurate graph recovery, whereas excessive regularisation degrades performance.

- **Causal annealing schedules (Appendix E.2.1):** Linear causal annealing achieves performance comparable to hard annealing across ER settings, indicating robustness w.r.t the annealing strategy.

- **Additional experiment on Sachs protein–signaling dataset (early results(Appendix F)):** An early evaluation is performed on the Sachs protein–signaling dataset, the model recovers meaningful directed relationships consistent with known biological pathways, while bidirected edges are interpreted qualitatively due to the absence of ground-truth confounding annotations.

- **Shallow vs. complex confounding and worst-case analysis (Appendix G):** The analysis indicates that the model captures shallow and structured confounding, including variables with both directed and bidirected roles (e.g., FC $x_4$). Performance degradation may arise under extremely dense overlapping confounding, while directed edge recovery remains robust and bidirected estimation degrades gracefully.

