# OpenReview forum: "Generative Causal Structure Learning with Dual Latent Spaces and Annealing"
_TMLR — Accepted by TMLR_

### Review · Reviewer_Bhj7 · 2025-11-03

**Summary Of Contributions:**

Authors of this paper proposed a novel generative method for causal structure learning in the presence of unobserved confounders. The proposed model is built on variational autoencoder with dual latent spaces to represent directed cause-effect relations and the bidirected unobserved confounded relations. A causal mixed graph loss and a novel causal annealing strategy are introduced to learn meaningful causal structures. Experiments on synthetic and real-world datasets demonstrating the effectiveness of the proposed model for causal structure learning in the presence of unobserved confounders. However, the proposed causal mixed graph loss is not well defined and explained, so it is difficult to justify the correctness of the proposed model. The experiments could be significantly improved by reporting robust convincing results and providing proper guidelines to setup model hyperparameters.

**Audience:**

Yes

**Audience Explanation:**

The disentangle of two different graphs based on VAE may be interested for a small group of individuals.

**Claims And Evidence:**

No

**Claims Explanation:**

There are several critical concerns. First, the proposed model is not well presented, especially the proposed causal mixed graph loss. This makes the justification of the proposed model difficult. Second, the experimental results are not reported in a convincing way, such as the best result of 5 runs is reported, and the analyses are performed on the best results. Third, the proposed model is a combination of many factors, but their contributions are not properly explored. Since there lacks the guidelines to setup so many hyperparameters in the proposed model, the usability of the model may be limited.

**Requested Changes:**

The loss function in (3) is a combination of many items. However, these loss functions are not properly explained or even defined. For example, what is the reconstruction loss in VAE? Authors need to explain why minimizing (7)-(8) can achieve acyclicity and bow-free. Notice that the losses of symmetry-asymmetry and sparsity are not defined. And the sparsity on A_B is missing. It is recommended to clarify these loss terms clearly and accurately.

What is the adjacency matrix estimation in step 17 of Algorithm 1?

In section 4, 4000 samples for FC are sampled, but it is unclear for ER.

The results in section 5 are reported using the best run over 5 repeated runs. According to Table 5 and Table 6, the best result is much better than the mean of 5 runs. In addition, the repeated experiments for ER are done for randomly sampled graph structure, while the varied number of samples can also have impact on the quality of the learned causal structure. In real application, the ground truth graph is unknown, so it is impossible to select the best run. The recommendation is to report the mean+std for all compared methods in Table 1. The case study could be done for the best and the worst runs to see how different they are.

Ablation study in Table 2 shows that all items of the proposed causal loss contribute to the final output. However, the study was only performed by excluding one item each time. This is not helpful to set so many hyperparameters in (3) as did in Table 7. Notice that the hyperparameters are not constant for all three datasets, and some hyperparameters are not included in the table such as $\lambda_{KL}$ and $\lambda_{sparse}$. It seems that the sparsity hyperparameter should be more important to the final graph structures, but it is not investigated. Authors need to provide proper parameter sensitivity analysis to provide guidelines for users.

Causal annealing experiment shown in Table 3 was done for comparing hard causal annealing and without annealing. However, the experiments are done for linear interpolation. Authors are recommended to add one with linear case. In addition, the CTE and linear transition start epoch $e_t$ will have impact on the final output. So, the ablation study over these hyperparameters is interested.

---

> ### Author Response · Authors · 2025-12-03
> **Addressing the reviewer comments (Reviewer Bhj7)**
>
> We thank the reviewer for the detailed and insightful comments. We address each point below:
>
> 1. Loss terms not defined: Eq. (3) is fully expanded: reconstruction loss, symmetry/asymmetry, and sparsity for both A_D and A_B are now explicitly defined.
>
> 2.Explanation of eq (7)- (8 ): We added explanations in the revised manuscript clarifying why minimizing eqs. (7)–(8) enforces acyclicity and bow-free structure (page 5).
>
> 3. Algorithm 1 clarification:  Step 17 now explicitly state how the adjacency matrices are obtained from the structure-aware latent representations.
>
> 4. ER sample size: Clarified that ER datasets also use 4000 samples.
>
> 5. Reporting best run vs. mean/std:  Table 1 now reports mean ± std over five runs; In addition, we include an adaptive-thresholding analysis in Appendix B.2 as a data-driven alternative when the ground truth is unavailable. We retain the “best run” as a case study in Appendix C (as was previously included ) to illustrate variability in graph recovery quality.
>
> 6. Hyperparameter sensitivity analysis: Detailed analysis added in Appendix E. We added bow-sensitivity (Fig. 7), sparsity-sensitivity (Table 10), and a detailed guidelines in Appendix E. These clarifications describe the roles of  λ_sparse, λ_entropy, λ_bow, λ_cycle, as well as the effects of over-sparsification, and observed behaviors during training.
>
> 7.Causal annealing (linear case): An initial linear-annealing study is included in Appendix F, comparing hard and linear schedules and effects of CTE and transition epochs.

---

### Review · Reviewer_dbKT · 2025-11-03

**Summary Of Contributions:**

The paper proposes a VAE-based method G-ADMG-CL for disentangles unobserved confounders. The core contribution lies in learning dual latent spaces, which seperately model the directed adjacency matrix, and the bidirected adjacency matrix. The method stabilizes training through a set of causally aware objective functions and a causal-annealing training strategy.

Strengths:
1) The authors ground their assumptions and training constraints in established identifiability results, ensuring a solid theoretical foundation.
2) The ablation studies and thorough comparative experiments enhance the credibility of the results.

Weakness：
1) The implementation is limited to small-scale variables and semi-synthetic data. Although the authors emphasize that the method benefits downstream causal-effect estimation tasks, the experimental conclusions do not fully demonstrate this advantage.
2) There is no sufficient theoretical proof establishing the consistency between the dual latent spaces disentangled from data and the observed/unobserved causal graphs.

**Audience:**

Yes

**Audience Explanation:**

Causal structure learning in the presence of latent confounders is an important research problem. The authors introduce a variety of causally-aware objective functions and, by combining them with a VAE framework, greatly enhance the flexibility of the work, offering rich inspiration to the causal community.

**Broader Impact Concerns:**

I do not see any concern on the ethical implication.

**Claims And Evidence:**

Yes

**Claims Explanation:**

The supporting evidence is strong, and the experiments are thorough. The ablation studies and thorough comparative experiments enhance the credibility of the results.

**Requested Changes:**

1) Add identifiability guarantees and a comprehensive failure-mode analysis: clarify the conditions under which A_D and A_B are recoverable (addressing Weakness 2), identify which edge types are most error-prone, and specify when A_B is likely to be misclassified as A_D. Such supplements will strengthen theoretical confidence and guide practical deployment.

2) Thresholding sensitivity analysis required: Report how A_D and A_B F1-scores vary with different binarization thresholds, clarify whether per-dataset tuning is necessary, and compare a single global threshold against an adaptive one. This ensures that reported performance is reliable.

3) The current experimental results can demonstrate the effectiveness of the work; however, the evidence regarding its impact in real-world scenarios, as claimed in the paper, remains somewhat weak.Expand real-world and down-stream validation: Beyond IHDP, include at least one additional real-world dataset with latent confounding and provide structure-level metrics. For down-stream validation, synthetic or real-world causal effect estimation is necessary.

---

> ### Author Response · Authors · 2025-12-03
> **Addressing the reviewer comments (Reviewer dbKT)**
>
> Thank you for your helpful and detailed feedback. We address each point below:
>
> 1. Identifiability & failure modes: Appendix D now details identifiability limits for ADMGs.
>     Appendix H adds failure-mode analysis covering: weak effects, near-deterministic confounding, overlapping confounders, and misclassification scenarios for A_B as A_D. Worst-case ER(12,50,10) analysis included.
>
> 2.Threshold-sensitivity We added: (i) full threshold–F1 curves for A_D and A_B (Appendix B.1), (ii) an adaptive thresholding method, a data driven approach (Appendix B.2) (iii) comparison of best vs. adaptive thresholds. FC results remain stable under all strategies.
>
> 3.Real-world validation: Appendix G adds the Sachs dataset with directed/bidirected edge evaluation and visualizations.

---

### Review · Reviewer_2qMB · 2025-11-03

**Summary Of Contributions:**

Thank you for working towards a solution to the hard problem of causality within AI.
In the following, I try to provide an honest, respectful and most importantly helpful review such that this work can grow to be a great contribution to our community.

The present manuscript provides an effort in using a VAE approach to handle the task of learning a causal structure from data.
The data is generated by a Pearlian SCM and we are only given a causally insufficient set of variables i.e., hidden confounders are present in the actual data generating process.
The authors model the structural relationships between any pair of variables by splitting the task at hand into a directed and bi-directed graph respectively.
The authors model this split via two latent layers in the VAE that are being optimized in tandem.
The authors make use of various learning strategies (mainly, structural constraints and annealing).
The authors present experiments two-fold: 1) how well the method is able to recover the correct structure(s), 2) how well the learned structure(s) perform in downstream causal inference.

The strengths of the paper IMHO are:
* A great research problem
* Ablating across various learning techniques
* Detailled technical description for reproduction

The weakness of the paper IMHO are:
* No guarantee for causality in the learned model
* Triviality: "Shallow" hidden confounder
* Missing scholarly input (insufficient related work)
* Weak experimental evidence
* Error-prone / Unpolished presentation
* Missing code base for reproduction

**Audience:**

Yes

**Audience Explanation:**

Since the TMLR community is concerned with not just causality and causality for AI but more importantly general learning approaches, in particular also for structure learning, this work speaks to a broader audience.

**Broader Impact Concerns:**

I've no concerns of any ethical nature regarding this work, whereas the potential impact of a success in this particular domain of research would have far reaching implications in all of science and industry.

**Claims And Evidence:**

No

**Claims Explanation:**

Regarding the strengths:
* A great research problem: since causality for AI is far from solved, yet its inferences are undeniably necessary for any sort of actual AI, any progress towards the matter, however miniscule, would be important.
* Ablating across various learning techniques: since any of the given techniques can play a crucial role in promoting any given method to outperform the competition in any given recovery metric.
* Detailled technical description: since reproducibility is a pillar of science and especially nowadays with growing complexities in systems it is in particular important to know how to reproduce a given result (I've not reproduced any of the results and also I could not find any code that was provided, however, the descriptions at least do seem precise enough for reproduction).

Regarding the weaknesses:
* No guarantee for causality: as with other continuous-optimization approaches to structure learning (e.g. NOTEARS) we focus on learning DAGs, and in the case of the present manuscript, ADMGs. While these structures are used in causality and also the data is assumed to stem from a corresponding SCM, this family of methods does not guarantee learning the structure of said SCM. No term in the optimization can actually enforce that and the model itself does not maintain an ability to proclaim said result.
* "Shallow" hidden confounder: as per Reichenbach's common cause principle, we know that if there is correlation between two variables, then there must be causation present either in the form of direct causation between said two variables (in which case correlation implies causation) or in the form of a third variable, a hidden confounder, causing both variables (in which case correlation does not imply causation). The authors use a "bow-free" constraint. The bow referring to the idea that the structure of three variables resembles a bow, when there is a hidden confounder but also a direct causation between the two named variables. Now, if we have a bow-free assumption, then necessarily it must be true that the correlation that we observe between the two given variables stems from a hidden confounder, therefore, rendering this result trivial.
* Missing scholarly input: the referenced related work is insufficient in covering the existing literature, specifically also with similar or adjacent VAE approaches in consideration. What comes to mind immediately are for instance following three works on causality and VAE (some also do CSL, whereas others focus on inference - the authors downstream task): (1) [Yang et al. 2021](https://openaccess.thecvf.com/content/CVPR2021/papers/Yang_CausalVAE_Disentangled_Representation_Learning_via_Neural_Structural_Causal_Models_CVPR_2021_paper.pdf), (2) [Leeb et al. 2023](https://arxiv.org/abs/2106.16091), and (3) [Matej Zečević et al. 2021](https://arxiv.org/abs/2109.04173)
* Insufficient experimental evaluation: the deployed testing ground is well-known in the causal community and IMHO generally considered more as a sanity-check rather than an actual real world relevant assessment. Given that the presented work is in particular keen on beating any state of the art approach, this point actually becomes a worthwhile weakness to discuss.
* Error-prone / Unpolished presentation: the present manuscript suffers from several language and typesetting mistakes. Furthermore, the complete style of presentation is rather reminiscent of a "technical report" rather than a scientific paper.
* Missing code base for reproduction: it is crucial for a scientific community to be able to independently evaluate any given result of technical form, in particular, when the key contribution is of technical nature. Ideally, the VAE approach should be plug-and-play available to anyone (for now, the reviewers)

**Requested Changes:**

Going over the weaknesses one by one:
* No guarantee for causality: this is a fundamental thing and I cannot expect the authors to resolve this within a revision. However, we need to provide a disclaimer on this and more than just a single sentence since this is a sensitive and important topic.
* "Shallow" hidden confounder: this is my biggest concern since to me the "learning about unobserved confounders" aspect of this structure learning approach is proclaimed (or should be) its biggest selling point. This is the actual core research problem that is supposed to be solved, I cannot give any proposed answer to this.
* Missing scholarly input: please once again do a thorough scholarly search, make use of the references I've provided and dangle within their citations trees to find more relevant work more easily and extend the existing discussion, especially w.r.t. (dis-)similarities.
* Insufficient experimental evaluation: within the causality community we make use of more complex settings e.g. robot environments (for example the MPI tri-finger environment: https://webdav.tuebingen.mpg.de/trifinger/), test within these simulators.
* Error-prone / Unpolished presentation: ideally let internal, independent reviewers pass over the paper to find mistakes that you as authors are unable to spot anymore since you have been working so much on this. I'd much more prefer this over just an AI pass, however, even that should've resolved the mistakes that I found within the present version.
* Missing code base for reproduction: prepare the code for sufficient online presentation (i.e., instruction on how to run and dependency lists) and post it online (anonymously) and provide the URL.

---

> ### Author Response · Authors · 2025-12-03
> **The revised manuscript addressing the reviewer comments (Reviewer 2qMB) .**
>
> Thank you for your constructive feedback. We address each point below:
>
> 1.No guarantee for causality: We added an expanded disclaimer in Appendix D (“Identifiability Assumptions and Causality Disclaimer”). It clarifies that: (i) recovery of the true SCM is not guaranteed; (ii) learned ADMGs represent plausible hypotheses; (iii) latent-confounder placement is non-identifiable; (iv) Markov equivalence creates ambiguity.
>
> 2.Shallow vs. complex confounding: A dedicated discussion is added in Appendix H(Discussion and Future work), explaining when shallow confounders are recoverable, and why dense/overlapping confounding reduces A_B accuracy. Worst-case bidirected results and failure modes are also included.
>
> 3.Missing scholarly input: We added the suggested works: Yang et al. and Leeb et al. (after Algorithm 1), Zečević et al. (Related Work), clarifying our model’s relation to existing approaches.
>
> 4.Insufficient evaluation: We strengthened experiments by adding: (i) Sachs real-world dataset (Appendix G), with structure-level evaluation and thresholding, (ii) complex-confounding analysis (Appendix H). Robotic simulators (TriFinger, Meta-World) are now explicitly listed in Future Work.
>
> 5.Presentation issues: We revised notation, fixed inconsistencies, and improved clarity throughout.
>
> 6.Reproducibility / code: All algorithms and hyperparameters are now fully specified. An anonymized OSF project is created: https://osf.io/am6bw/ . In addition, the runnable code has been uploaded to the OSF project to ensure full reproducibility.

---

### Decision · Action_Editor_HsFQ · 2025-12-30

**Recommendation:** Accept with minor revision

**Additional Comments:**

Overall reviewers think the proposed algorithm is practically useful. However many of them pointed out the lack of theoretical support, e.g., identifiability under suitable assumptions. Also reviewers suggested additional ablation studies.

I recommend in the camera ready the authors should revise the manuscript based on reviewers' suggestions. Make sure the narrative is clear (estimation algorithm instead of achieving identifiability), and provide a good summary regarding the experimental results (after adding the new results as requested by the reviewers).

**Audience:**

Yes

**Audience Explanation:**

Researchers interested in representation learning and causal discovery.

**Claims And Evidence:**

Yes

**Claims Explanation:**

This paper considers estimating Acyclic Directed Mixed Graphs (ADMGs) from data. The algorithm is based on VAEs plus novel regularisation terms designed based on the causal constraints on the graphs.

Experiments are conducted on simulated graphs up to ~10 nodes, and a semi-synthetic setting using simulated outcomes from a real-world dataset (IHDP). Baseline methods include classical causal discovery methods (like FCI) and previous neural network methods (e.g., flow-based ADMG estimators). The proposed approach is reported to be the best for many of the experiments when compared with baselines. Ablation studies mainly focused on the components in the proposed regularisation loss.